# Attribution of recent increases in atmospheric methane through 3-D inverse modelling

Joe McNorton[1,2], Chris Wilson[1,3], Manuel Gloor[4], Rob J. Parker[3,5], Hartmut Boesch[3,5], Wuhu Feng[1,6], Ryan Hossaini[7], Martyn P. Chipperfield[1,3]

[1]School of Earth and Environment, University of Leeds, Leeds, UK.
[2]Research Department, European Centre for Medium-Range Weather Forecasts, Reading, UK.
[3]National Centre for Earth Observation, University of Leeds, Leeds, UK.
[4]School of Geography, University of Leeds, Leeds, UK.
[5]Earth Observation Science Group, Department of Physics and Astronomy, University of Leicester, Leicester, UK.
[6]National Centre for Atmospheric Science, University of Leeds, Leeds, UK.
[7]Lancaster Environment Centre, Lancaster University, Lancaster UK.

*Correspondence to*: Joe McNorton (Joe.McNorton@ecmwf.int)

**Abstract.**

The atmospheric methane ($CH_4$) growth rate has varied considerably in recent decades. Unexplained renewed growth after 2006 followed seven years of stagnation and coincided with an isotopic trend toward $CH_4$ more depleted in $^{13}C$, suggesting changes in sources and/or sinks. Using surface observations of both $CH_4$ and the relative change of isotopologue ratio ($\delta^{13}CH_4$) to constrain a global 3-D chemical transport model (CTM), we have performed a synthesis inversion for source and sink attribution. Our method extends on previous studies by providing monthly and regional attribution of emissions from 6 different sectors and changes in atmospheric sinks for the extended 2003-2015 period. Regional evaluation of the model $CH_4$ tracer with independent column observations from the Greenhouse gases Observing SATellite (GOSAT) shows improved performance when using posterior fluxes (R = 0.94-0.96, RMSE = 8.3-16.5 ppb), relative to prior fluxes (R = 0.60-0.92, RMSE = 48.6-64.6 ppb). Further independent validation with data from the Total Carbon Column Observing Network (TCCON) shows a similar improvement in the posterior fluxes (R = 0.87, RMSE = 18.8 ppb) compared to the prior (R = 0.69, RMSE = 55.9 ppb). Based on these improved posterior fluxes, the inversion results suggest the most likely cause of the renewed methane growth is a post-2007 1.8±0.4% decrease in mean OH, a 12.9±2.7% increase in energy sector emissions, mainly from Africa/Middle East and Southern Asia/Oceania, and a 2.6±1.8% increase in wetland emissions, mainly from Northern Eurasia. The posterior wetland flux increases are in general agreement with bottom-up estimates, but the energy sector growth is greater than estimated by bottom-up methods. The model results are consistent across a range of sensitivity analyses. When forced to assume a constant (annually repeating) OH distribution, the inversion requires a greater increase in energy sector (13.6±2.7%) and wetland (3.6±1.8%) emissions and an 11.5±3.8% decrease in biomass burning emissions. Assuming no prior trend in sources and sinks slightly reduces the posterior growth rate in energy sector and wetland emissions, and further increases the magnitude of the negative OH trend. We find that possible tropospheric Cl variations do not to influence $\delta^{13}CH_4$ and $CH_4$ trends, although we suggest further work on Cl variability is required to fully diagnose this contribution. While the study

provides quantitative insight into possible emissions variations which may explain the observed trends, uncertainty in prior source and sink estimates and a paucity of $\delta^{13}CH_4$ observations limit the robustness of the posterior estimates.

## 1 Introduction

The atmospheric concentration of methane ($CH_4$) has been increasing globally since 2007, following a slowdown in growth from 1999 to 2006 (Dlugokencky *et al.*, 2017). The onset of the observed increase in $CH_4$ coincides with an isotopic trend to lighter $CH_4$, more depleted in $^{13}C$ (Nisbet *et al.*, 2014). The $^{13}CH_4$:$^{12}CH_4$ ratio (denoted by the $\delta^{13}CH_4$ value) is controlled by both the isotopic signatures of the sources and the isotopic fractionation associated with atmospheric $CH_4$ sinks. Broadly speaking, the emission types can be categorised into the relatively light biogenics (~-62‰), heavier fossil fuels (~-44‰) and the even heavier biomass burning emissions (~-22‰) (Schwietzke *et al.*, 2016), resulting in a total isotopic source signature of between -51‰ and -53‰. Isotopic fractionation in the atmosphere by the reaction with the hydroxyl (OH) radical and chlorine (Cl) atoms enriches $^{13}CH_4$, causing a background atmospheric $\delta^{13}CH_4$ of ~-47‰.

Previous studies have used simple global box-models for source and sink attribution of recent atmospheric $CH_4$ trends, with contradictory findings. Nisbet *et al*. (2014; 2016) and Schaefer *et al.* (2016) suggested that either increased wetland or agricultural emissions were the likely cause while Rigby *et al* (2017) and Turner *et al.* (2017) found the most likely explanation to be a decreased global mean OH concentration. The latter two studies emphasised that the problem is not very well constrained by existing data and as a result could not discard the hypothesis that OH is not changing. These approaches are able to isolate the three emission categories noted above, and sometimes sink terms. Specific attribution, for example between wetlands and agricultural emission changes, requires spatial representation of both $CH_4$ and $\delta^{13}CH_4$. The box-model approach provides little or no information of spatial variation in posterior emission estimates, preventing regional attribution. Rice *et al*. (2016) performed a 3-D chemical transport model (CTM) inversion using $CH_4$ and isotopologue measurements over the period 1984 to 2009. They found a 24 Tg yr$^{-1}$ increase in fugitive fossil fuel emissions between 1984 and 2009, most of which occurred after 2000. The time trend in their inversion appeared similar to their prior emission estimates. Although they used a 3-D CTM the posterior emissions were calculated globally and not regionally. Furthermore, their study did not focus on the possible role of OH variations and did not consider inversions after 2009, so only captured two years of the continued post-2007 growth.

Here we perform a synthesis inversion using the TOMCAT 3-D CTM, building on previous work (Bousquet *et al.*, 2006; Bousquet *et al.*, 2011; Rigby *et al.*, 2012; Schwietzke *et al.*, 2016; Rice *et al*., 2016) and using surface measurements of both $CH_4$ (Dlugokencky *et al.*, 2017) and $\delta^{13}CH_4$ (White *et al.*, 2017). The synthesis inversion technique uses the forward 3-D CTM to optimise monthly $CH_4$ emissions over relatively large regions and for multiple source sectors. This spatial resolution is not present in existing box model inversions. We investigate regional source contributions and the roles of tropospheric OH and

Cl in the recent growth of CH₄. From this we derive possible source and sink changes between 2003 and 2015 which best fit the observations.

## 2 Models and Observations

### 2.1 Chemical Transport Model

### 2.1.1 Forward model

The TOMCAT global CTM (Chipperfield *et al.*, 2006) has previously been widely used to simulate CH₄ trends and has been evaluated against observations (e.g. Patra *et al.*, 2011; Wilson *et al.,* 2016; Parker *et al.*, 2018). Here we base our synthesis inversions on TOMCAT simulations at $2.8° \times 2.8°$ resolution with 60 vertical levels from the surface to 60 km for 2003-2015. The simulations used meteorological forcing data from the 6-hourly European Centre for Medium-Range Weather Forecasts

ERA-Interim reanalyses (Dee *et al.*, 2011). The model was spun up from a 1977 initialisation field before the mean global CH₄ and $\delta^{13}CH_4$ were rescaled to match NOAA observations in January 2002. A one-year inversion spin-up was then performed for 2002, to optimise the 3-D CH₄ and $\delta^{13}CH_4$ concentration fields relative to observations and the results shown here begin in January 2003.

Monthly varying methane emissions from McNorton *et al.* (2016a) were updated using revisions based on Schwietzke *et al.* (2016), which increased fossil fuel emissions and decreased biogenic emissions compared to the estimates in Saunois *et al.* (2016). OH and stratospheric CH₄ loss fields were taken from McNorton *et al.* (2016b) and a TOMCAT-derived tropospheric Cl loss field (Hossaini *et al*., 2016) was applied for the first time in our model.

Emissions were grouped into individual tracers for agriculture (excluding rice), biomass burning, energy, rice, waste, wetlands and 'supplementary', made up of the remaining sources (geological, hydrates, oceans and termites). Each source type, excluding 'supplementary', was then sub-divided into five geographic regions; North America (NA), Northern Eurasia (EA), South America (SA), Africa and Middle East (AM), and South Asia and Oceania (AO) (see Figure 7). These regions were chosen by grouping existing Transcom regions (DeFries *et al*., 1994) and considering both socio-economic and biome

similarities. The aggregation of regions by combining both socio-economic and biome considerations is somewhat subjective and differing aggregations may influence synthesis inversion results (Kaminski *et al*., 2001), which represents a limitation of the inversion method used in this study. For example, there are socio-economic differences within the EA region, which may result in differing trends in anthropogenic emissions that cannot be resolved using the chosen aggregation; however, biome similarities inside the region mean that the aggregation is appropriate for natural fluxes. We split the Asian regions to derive

suitable posterior estimates for e.g. boreal and temperate wetlands (EO) and tropical wetlands (AO), although this may affect the posterior energy sector emissions for these regions. Increasing the number of regions would decrease the influence of the

aggregation method; however, the computational cost of simulating the tracers required for the synthesis inversion for different sectors and for 12 months effectively limits the number of possible regions that we could use to five.

To assess monthly emission variability, individual tracers were simulated for each month of the year, excluding 'supplementary' emissions, which were simulated annually. Emissions were further split into separate $^{12}CH_4$ and $^{13}CH_4$ tracers using isotopic source signatures taken from Schwietzke *et al.*, (2016) (Table 1), resulting in 6 source types over 5 regions for 12 months and 2 isotopologues, with an additional 5 regions for 'supplementary' sources (a total of 730 tracers). Kinetic fractionation (Table 1) was accounted for in the atmospheric loss of $^{13}CH_4$. The simulated tracers were then used to calculate $CH_4$ concentration and $\delta^{13}CH_4$ values. To investigate sensitivity to OH and Cl variations, three additional simulations were performed, a control, an OH-enhanced simulation (1% increase) and a tropospheric Cl-enhanced simulation (1% increase). Any feedback, on the $CH_4$ term within the loss rate, from the small adjustments made (1%) is assumed to be negligible.

### 2.1.2 Synthesis inversion

Our global synthesis inversions build on techniques used in Bousquet *et al.*, (2006), Bergamaschi *et al.*, (2007) and Rigby *et al.*, (2012). Prior estimates of sources and sinks, uncertainty estimates, and observations of both $CH_4$ and $\delta^{13}CH_4$ were used to quantify posterior estimates of sources and sinks. Posterior estimates were then used in a second forward simulation for the same year, which provided an initialisation field for the subsequent year. The inversion method is limited by the assumption that isotopic source signatures are known.

For the inversion including OH concentrations in the state vector we consider the total simulated $CH_4$ mixing ratio ($\varphi$) and the $\delta^{13}CH_4$ value ($\psi$) at time, $t$, at each measurement location, $l$. These are described as a linear combination of contributions from $n_{reg}$ emission regions separated into $n_{month}$ months and $n_{source}$ emission sectors, loss due to OH, fractionation due to OH, the initial mixing ratio at the location, $\varphi_{ini}$, and the initial $\delta^{13}CH_4$ value at the location, $\psi_{ini}$:

$$\varphi(\boldsymbol{x}, l, t) = \sum_{s=1}^{n_{source}} \sum_{i=1}^{n_{reg}} \sum_{m=1}^{n_{month}} x_{i,m,s} \frac{\Delta\varphi}{\Delta x_{i,m,s}}(l,t) + x_{OH} \frac{\Delta\varphi}{\Delta x_{OH}}(l,t) + x_{ini}\varphi_{ini}(l) \qquad (1)$$

$$\psi(\boldsymbol{x}, l, t) = \sum_{s=1}^{n_{source}} \sum_{i=1}^{n_{reg}} \sum_{m=1}^{n_{month}} x_{i,m,s} \frac{\Delta\psi}{\Delta x_{i,m,s}}(l,t) + x_{OH} \frac{\Delta\psi}{\Delta x_{OH}}(l,t) + \psi_{ini}(l) \qquad (2)$$

Note that we use $\Delta$ here to represent change, to avoid confusion with the isotopologue $\delta^{13}CH_4$. Basis functions $\frac{\Delta\varphi}{\Delta x_{i,m,s}}$ and $\frac{\Delta\psi}{\Delta x_{i,m,s}}$ are sensitivities of atmospheric $CH_4$ and $\delta^{13}CH_4$ at a particular time and location to an emission of 1 Tg of $CH_4$ from a region $i$ during a particular month $m$, for an emission sector $s$. Each $x_{i,m,s}$ is a scaling factor applied to the contribution from each

basis function, and is initially set equal to the prior value of the emission. Similarly, $\frac{\Delta \varphi}{\Delta x_{OH}}$ and $\frac{\Delta \psi}{\Delta x_{OH}}$ are the sensitivities of the mixing ratio and $\delta^{13}CH_4$ at a measurement location to a change in the global OH concentration, linearised around the prior, and $x_{OH}$ is initially set to be the prior OH concentration. $x_{ini}$ is a dimensionless scaling factor initially set to be 1. Although the emissions in each region and source type are split into $^{12}CH_4$ and $^{13}CH_4$, the relative emissions of each isotopologue from

each region for each source type are not included as separate basis functions. The 'state vector' $x$ comprises of the individual emission scaling factors $x_{i,m,s}$, for all $i$, $m$ and $s$, along with $x_{OH}$ and $x_{ini}$. Sensitivity experiments performed for tropospheric Cl follow the same formulation with Cl terms replacing OH terms.

Varying atmospheric $CH_4$ concentrations in the inversions should in principle result in a non-linear feedback on OH
concentration. This feedback is not accounted for in the offline OH field used in our inversion. To resolve this, an online OH field could be used with an iterative minimization of the cost function. However, Bousquet *et al.* (2011) found that the small variation in $CH_4$ concentration between the prior and posterior had a negligible influence on OH concentration.

The model OH is constrained by $CH_4$ and $\delta^{13}CH_4$ but not by other species, such as methyl-chloroform (MCF). MCF was
excluded because of uncertainty in emissions and a diminishing concentration (<5 ppt), particularly during the later period of the study (Liang *et al.*, 2017). Due to the large uncertainty relative to the observed MCF concentrations in this period, including the extra species within the inversion would not add any extra constraint on the global OH concentration.

Independent inversions (INV-FULL) were performed for each year from 2003 to 2015. Initial conditions for each year are
provided by a forward simulation for the previous year driven by derived posterior emissions and loss rates, with 2003 initial conditions taken from a 2002 spin-up inversion. To quantify the optimisation of the flux terms in each region and the sink term, we calculate the cost function, $J$:

$$J(x) = \frac{1}{2}(x - x^b)^T B^{-1}(x - x^b) + \frac{1}{2}(y - Gx)^T R^{-1}(y - Gx) \quad (3)$$

The value of this 'cost function' is dependent on the value of the state vector $x$. The vector $y$ contains the observations. $x^b$ is the *a priori* estimate of $x$, and $B$ is the error covariance matrix containing the uncertainties placed on the prior estimates, and the covariances between these uncertainties. $G$ is the sensitivity matrix, which maps $x$ onto the observations, and contains an array made up of the basis functions, $\frac{\Delta \chi}{\Delta x}$ and $\frac{\Delta \psi}{\Delta x}$ used in Eq. (1) and (2). $R$ is the diagonal error covariance matrix for the
observations and model error.

The minimum of the cost function, which indicates the optimal source/sink scaling, is found using (Tarantola and Valette, 1982):

$$x^a = x^b + [G^T R^{-1} G + B^{-1}]^{-1} G^T R^{-1}[y - Gx^b] \qquad (4)$$

where $x^a$ is the optimised set of scaling factors which minimise the value of $J$.

The posteriori error covariance matrix $A$ is calculated from:

$$A = [G^T R^{-1} G + B^{-1}]^{-1} \qquad (5)$$

The initial prior uncertainty of each source within each region was set to 50%, based on uncertainties given by Kirschke *et al.*, (2013). We assume that increased uncertainty in sources with large interannual variability is offset by those sources having top-down (biomass burning) or process based (wetlands) interannually varying emissions in our simulations. We assumed small variability in energy sector emissions so assigned a 1-month offset correlation of 0.5, we have not assigned correlations between regions or months in the other prior emissions due to a lack of information. Global annual OH and Cl are assumed to
have an uncertainty of 2%; for OH this is based on estimated interannual variability (Montzka *et al.*, 2011). The impact of varying these uncertainties was investigated. Observational uncertainties were set at 10 ppb for $CH_4$ and 0.1‰ for $\delta^{13}CH_4$; the increase from the documented uncertainties is to represent model transport uncertainty that would otherwise only be resolved by emission changes. The magnitude of model transport will vary between different sites; however, as an estimate here we assume all uncertainties to be equal. By separating the inversion into 12 month intervals the emissions from the previous year
are not considered in the inversion for the current year. As a result, December emissions are constrained by fewer observations than January emissions. The influence of this on the posterior error is investigated in section 3.6.

To investigate the effect of including $\delta^{13}CH_4$ observations we performed a separate inversion (INV-CH4) using only $CH_4$ observations. The difference between the inversions indicates the additional information supplied by the inclusion of $\delta^{13}CH_4$.
Additional sensitivity experiments were also performed, 9 with varying prior uncertainties and an additional one with no prior trend in annual emissions, to investigate the robustness of the identified trends from the main inversion.

## 2.2 CH₄ and δ¹³CH₄ observations

Monthly mean measurements of $CH_4$ were taken from 21 National Oceanographic and Atmospheric Administration/Earth System Research Laboratory (NOAA/ESRL) air sampling sites (Dlugokencky *et al.*, 2017) from 2003 to 2015, where available.
Measurements of $\delta^{13}CH_4$ were taken from 11 NOAA sampling sites and analysed by the Institute of Arctic and Alpine Research (INSTAAR) (White *et al.*, 2017) for the same period (see Table 2). An equal weighting is applied to each monthly mean measurement and potential cross correlations from neighbouring time steps and spatially nearby sites are not considered.

Column-averaged CH$_4$ (XCH$_4$) GOSAT satellite data provided by the University of Leicester were not included in the inversion but retained for independent validation of the inversion results (Parker *et al.*, 2015). GOSAT was omitted because measurements were only available from 2009, 6 years after the inversion began. The Total Carbon Column Observing Network (TCCON) XCH$_4$ data were used as validation but were considered too intermittent for use in the inversion (Wunch *et al.*,

2011). Finally, two surface observation sites, The High Altitude Global Climate Observation Center (HAGCOC) in Mexico and Cape Grim in Australia were also used for independent validation.

## 3 Results

### 3.1 Synthesis Inversion

Inversion results constrained by CH$_4$ and $\delta^{13}$CH$_4$ observations (INV-FULL) show, as expected, improved seasonal and

interannual monthly averaged posterior CH$_4$ and $\delta^{13}$CH$_4$ estimates when compared with assimilated surface observations (Figure 1). The correlation with observations (R) for CH$_4$ increases from an all-site average of 0.72 in the prior to 0.94 in the posterior, and for $\delta^{13}$CH$_4$ increases from 0.52 to 0.87. Similarly, the root-mean-square error (RMSE) decreases from 38.2 ppb to 9.7 ppb for CH$_4$ and from 0.25‰ to 0.09‰ for $\delta^{13}$CH$_4$. The prior model captures some of the initial 2007 CH$_4$ growth but fails to capture the sustained growth (Figure 1a). The bias in the prior, relative to both the posterior and observations, grows

throughout the simulation period. This results in a large bias at the end of the time period, which is evident in the large RMSE values (Figures 1 to 4). The prior also shows a slight decrease in $\delta^{13}$CH$_4$ since 2007, but the magnitude of this is smaller than observed (Figure 1b). The renewed growth of CH$_4$ and corresponding decrease in $\delta^{13}$CH$_4$ in 2007 are well captured in the inversion.

Inversion results constrained by CH$_4$ (INV-CH4), and not $\delta^{13}$CH$_4$, also accurately reproduce assimilated CH$_4$ observations (R = 0.93). INV-CH4 also shows some improved agreement with $\delta^{13}$CH$_4$ observations relative to the prior (R = 0.60), although values are overestimated in earlier years (2003-2008) (Figure 1b).

Validation of the model inversion using the independent, non-assimilated GOSAT data shows improved seasonal and

interannual representation of XCH$_4$ (Figure 2). The RMSE is reduced in all 5 regions with values ranging from 48.6 to 64.6 ppb in the prior to 8.3 to 16.5 ppb in the posterior, with values typically originating from a negative bias in the model. The correlation is increased in the inversion with R values ranging from 0.60 to 0.92 in the prior to 0.94 to 0.96 in the posterior. The trend is also better captured in the posterior in all 5 regions, although still underestimated in all regions, more so in EA (-1.3 ppb yr$^{-1}$) and AO (-1.1 ppb yr$^{-1}$). Both the prior and posterior biases are larger in the southern hemisphere, possibly as a

result of slow inter-hemispheric transport within the model, previously noted in Patra *et al.* (2011). Also contributing to this offset is an underestimation of southern hemisphere simulated atmospheric CH$_4$ growth rates in the prior model simulation (Figure 3).

We performed further validation using measurements from 9 non-assimilated TCCON sites with data available from at least 2009 (see Table 3). The results show improved model-data correlation at all 9 sites, with an increase in the all-site mean R value from 0.69 in the prior to 0.87 in the posterior (Figure 4). The RMSE is reduced at sites, with an all-site mean decrease from 55.9 ppb in the prior to 18.8 ppb in the posterior, further reductions would be expected if column observations were used in the inversion. Overall the inversions are found to improve model-data agreement when validated against the independent measurements from both GOSAT and TCCON. The resulting southern hemisphere offset in the posterior relative to GOSAT and TCCON suggests the posterior estimates represent a reasonable but not conclusive scenario for source/sink attribution. As only surface sites are assimilated, some inaccuracy in the representation of the total column is not surprising.

Two surface sites were omitted from the inversion and retained for independent validation, HAGCOC and Cape Grim. Cape Grim is a baseline station, ideal for comparing the background signal. Results at Cape Grim show improved model performance for both $CH_4$ and $\delta^{13}CH_4$, with respective RMSE decreases from 52.0 ppb and 0.2‰ in the prior to 12.3 ppb and 0.1‰ in the posterior and R value increases from 0.70 and 0.74 in the prior to 0.95 and 0.76 in the posterior. HAGCOC measurements are taken at high altitude (4464m), which potentially provides insight into the vertical profile of measured species. As with Cape Grim, HAGCOC shows posterior improvements in both $CH_4$ and $\delta^{13}CH_4$, with respective RMSE decreases from 70.5 ppb and 0.2‰ in the prior to 27.1 ppb and 0.08‰ in the posterior and R value increases from 0.41 and 0.46 in the prior to 0.62 and 0.74 in the posterior.

**3.2 Prior and Posterior Comparison**

The synthesis inversions, INV-FULL and INV-CH4, provide posterior regional changes in sources and global changes in OH (Figure 6). Relative to the prior, INV-FULL and INV-CH4 show an average OH decrease of 5% and 4%, respectively (Table 1). Results from INV-FULL show that globally agricultural (-13%), energy (-8%) and biomass burning (+7%) emissions undergo the largest relative average 2003-2015 posterior change compared to the prior (Table 1). Relative changes in rice, waste and wetlands are smaller (<3%). The posterior emission errors are between 5%-13% compared with the 50% prior error. Regionally (Figure 7), 2003-2015 average posterior energy sector emissions are increased, relative to the prior, by 9-33% in four regions (NA, SA, AM and AO), which is offset by a 37% decrease in EA. Notable posterior agricultural emission decreases occur in EA (-36%) and AO (-14%). Wetland emissions are increased beyond the posterior error range in NA (+24%) and EA (+44%) and decreased within the error range in SA (-7%), AM (-7%) and AO (-6%). In all regions posterior emission estimates for biomass burning, waste and rice are within, or close to, the error range compared with prior estimates (Table 4).

Globally, for the 2003-2015 period, derived posterior and prior emission estimates had average growth rates of 4.1±0.6 Tg yr$^{-2}$ and 4.0±0.2 Tg yr$^{-2}$, respectively. When considering only the renewed growth (2007-2015) the posterior growth rate of 5.7±0.8 Tg yr$^{-2}$ becomes noticeably larger than the prior (3.7±0.4 Tg yr$^{-2}$).

The seasonal range of the prior global wetland emissions (5.7 Tg month$^{-1}$) is underestimated compared to the posterior (13.8 Tg month$^{-1}$). The seasonal cycle in biomass burning emissions is largely unchanged between the prior and posterior. The seasonal amplitude in rice emissions also remains largely unchanged, although the seasonal peak occurs in August in the prior and July in the posterior (Figure 6).

### 3.3 Time Trends in Sources and Sinks

Average energy, waste and wetland emissions are increased post-2007 by 12.9±2.7% (19.0 Tg yr$^{-1}$), 5.7±1.6% (3.8 Tg yr$^{-1}$) and 2.6±1.8% (4.0 Tg yr$^{-1}$), respectively, relative to their 2003-2006 posterior values (Table 6). Regionally, the shift in post-2007 energy sector emissions mainly occurs in AM (+8.4 Tg yr$^{-1}$) and AO (+11.1 Tg yr$^{-1}$). Four out of five of the regions show a positive post-2007 shift in waste emissions of 0.4-1.4 Tg yr$^{-1}$, SA is the only region with a slight decrease (-0.03 Tg yr$^{-1}$). The small increase in wetland emissions since 2007 derived from the inversion, mainly from EA (3.4 Tg yr$^{-1}$), agrees well bottom-up estimates for wetland emission trends, for example the 3% increase found by McNorton *et al*. (2016a). The posterior shows a negative shift in posterior biomass burning emissions 11.8±6.4% (-2.9 Tg yr$^{-1}$) for the 2007-2015 period relative to 2003-2006, which is in partial agreement with the 3.7 Tg CH$_4$ yr$^{-1}$ decrease derived by Worden *et al*. (2017) for the 2008-2014 period relative to 2001-2007. This shift occurs in all five regions, with the largest decrease in AO (-1.2 Tg yr$^{-1}$). Overall the derived increase in energy sector, waste and wetland emissions coupled with the decrease in biomass burning emissions agree well with a recent budget review (Saunois *et al*. 2017).

The post-2007 posterior emission growth occurs mainly in the energy (3.4±1.0 Tg yr$^{-2}$) and wetland (1.4±1.0 Tg yr$^{-2}$) sectors. For the entire period most of the posterior energy sector growth occurred in AM (1.2 Tg yr$^{-2}$) and AO (1.5 Tg yr$^{-2}$), with a smaller proportion from NA (0.6 Tg yr$^{-2}$) and SA (0.2 Tg yr$^{-2}$) (Figure 7 and Table 5). The recent EDGAR v4.3.2 inventory (Janssens-Maenhout *et al.,* 2017) for energy sector emissions shows AM and AO growth of 1.0 Tg yr$^{-2}$ and 2.4 Tg yr$^{-2}$, respectively, for 2003-2012. These are, smaller than the 2.2 Tg yr$^{-2}$ and 3.1 Tg yr$^{-2}$ shown by our inversion for the same period. A majority of prior AM energy sector emissions originate from energy for buildings in Nigeria and Eastern Africa, fuel exploitation from the Middle East, the Niger Delta and South Africa, and pipelines in Western Africa, Algeria and The Middle East. The regional aggregation of fluxes in our inversion system prevents sub-regional attribution, as a result we are unable to diagnose more specific posterior spatial patterns, but our results suggest on a regional scale, emissions are underestimated in both magnitude and growth rate in the prior. For the AO energy sector, a majority of prior emissions, and therefore the posterior increases, originate from energy for buildings in India, China and South-East Asia, fuel exploitation in Eastern China, Japan, India, South East Asia and Eastern Australia, refineries in Northern India, Eastern China, Japan and Indonesia, and pipelines in India Eastern China, Eastern Australia and New Zealand. The growth in emissions in EA in EDGAR v4.3.2 for 2003-2012 (1.4 Tg yr$^{-2}$) is not seen in our inversion for the same region and period (-2.2 Tg yr$^{-2}$).

During the 2008-2012 period NA energy sector emissions were found to be 11.4 Tg yr$^{-1}$ (+66%) higher than the 2003-2015 (excluding 2008-2012) average, resulting in uncertainty in the NA growth rate (Figure 6). These findings are also present in INV-CH4, which shows an 11.8 Tg yr$^{-1}$ increase over the same period. This period of anomalously high emissions is not present in the prior and therefore, is due to the assimilated observations. These high emissions may be associated with oil or natural gas extraction (Helmig *et al.*, 2016). During periods of high NA energy sector emissions, the EA energy sector emissions are reduced and vice-versa, suggesting a possible dipole caused by the inversion. This suggests increased uncertainty in the derived EA and NA energy sector emissions, possibly due to a paucity of observations over these regions.

Posterior wetland emission estimates show a growth of 0.8 Tg yr$^{-2}$ for the 2003-2015 period, which increases to 1.4 Tg yr$^{-2}$ for the 2007-2015 period. A majority of this growth occurs in EA (+0.5 Tg yr$^{-2}$). The four remaining emission sectors all have a global annual change less than ±0.5 Tg yr$^{-2}$.

For the posterior time series, OH concentrations in INV-FULL and INV-CH4 are relatively constant throughout the period 2007-2015 (Figure 6) but relative to their 2003-2006 concentrations these values are smaller by 1.8±0.4% and 0.3±0.5%, respectively. The larger drop post-2007 in INV-FULL OH concentration, relative to INV-CH4, highlights the importance of including $\delta^{13}CH_4$ in the inversion. A decrease in OH as a contributor to the renewed growth agrees well with previous simple global box models (Rigby *et al.,* 2017; Turner *et al.*, 2017). The OH shift found here is smaller in magnitude than the -8% shift between 2004 and 2014 derived by Rigby *et al.*, (2017) and the -7% shift between 2003 and 2016 derived by Turner *et al*., (2017). The posterior OH error is reduced from the prior estimate of 2% to 1.8%, which, although a reduction, is similar to the modelled post-2007 OH decrease. The decrease in OH contributes to a decrease in $\delta^{13}CH_4$ and an increase in global $CH_4$. Section 3.5 details analysis of OH sensitivity.

### 3.4 Source and Sink Attribution

Analysis performed on our inversion results using the box model approach described by McNorton *et al.* (2016b) suggests that ~30% of the sustained $CH_4$ growth post-2007 can be explained by decreased OH, while ~60% and ~10% is attributed to increased energy sector and wetland emissions (Table 5). The shift in emissions between 2003-2006 and 2007-2015 is broadly consistent for each sector for three different inversions, INV_FULL, INV_CH4 and INV_FIXED (fixed annual emissions, see below) (Table 6). We investigated source and sink contribution to the negative $\delta^{13}CH_4$ trend using simple one box model analysis, outlined in the appendix of McNorton *et al*. (2016b), and posterior estimates from INV-FULL. Results show that post-2007 changes in energy sector (+0.15‰), biomass burning (-0.08‰), wetland (-0.05‰), waste sector (-0.03‰) and agricultural (-0.01‰) emissions, as well as OH (-0.12‰), contributed to the observed trend.

### 3.5 Sensitivity Tests

To test the robustness of the inversion to changes in prior error estimates we performed nine perturbation experiments (S1-S9). Monthly source errors were perturbed between 10% and 100%, and yearly OH errors from 0% to 10% (Figure 8 and Table 7). For small error perturbations, the inversion results do not change much relative to INV-FULL (Figure 8 and Table 8). However, when the emission errors are reduced from 50% to 10% (S4) the posterior energy emissions estimates deviate from the control (INV-FULL) inversion, with a mean bias of 60.5 Tg yr$^{-1}$. We consider these large ranges in posterior estimates to be an unrealistic representation of interannual variability in energy sector emissions (Figure 8), which suggests the model fails to provide reasonable posterior estimates when the prior emission error is set too low. For most cases of increased emission errors the OH change is similar to the control. However, for 100% emission errors (S6) the agricultural emissions are further reduced, from 82.8 Tg yr$^{-1}$ in the prior and 72.1 Tg yr$^{-1}$ in INV-FULL, to 64.1 Tg yr$^{-1}$. In this case OH is only reduced by 0.5% post-2007, relative to 2003-2006, compared to 1.8% in INV-FULL. This results in a smaller OH contribution to the post-2007 CH$_4$ growth.

For large or small OH errors (S3: 10%, S1: 1%) the posterior OH is decreased by 18% or 2%, respectively, compared to the prior OH. Assuming no change in OH (S9) post-2007 shifts in biomass burning, energy sector and wetland emissions relative to 2003-2006 are required to fit observations in the inversion. In this scenario biomass burning emissions decrease globally by -11.5±3.8% (-2.9 Tg yr$^{-1}$) and in AO by -16.1±17.9% (-1.2 Tg yr$^{-1}$). Energy sector emissions increase globally by 13.6±2.7% (+20.6 Tg yr$^{-1}$), in NA by 42.9±12.9% (+7.7 Tg yr$^{-1}$) and in AO by 36.7±5.1% (+12 Tg yr$^{-1}$). Wetland emissions increase globally by 3.6±1.8% (+5.8 Tg yr$^{-1}$). The sign and spatial distribution of these changes are similar to those seen in INV-FULL although the magnitude in post-2007 changes is typically increased in S9 (see Section 3.2), which is expected as the necessary increased growth rate is allocated more to emission changes when OH is assumed constant.

The sensitivity analyses highlight that the prior uncertainty can have a noticeable influence on the posterior estimates. In particular, the posterior OH is found to be sensitive to the prior error estimate, highlighting the importance of prior knowledge for future studies. This limits the accuracy of the magnitude of the posterior estimates. However, the spatial, temporal and sector specific relative post-2007 changes, compared to 2003-2006, remain broadly consistent between experiments. This shows a limitation in the comparison between prior and posterior sources/sinks but does not discount the importance of the results for trend detection between 2003 and 2015.

We performed a synthesis inversion with no prior trend in emissions or OH (INV-FIXED), using fixed 2003 emissions, to investigate the sensitivity of the inversion to prescribed prior trend information (Figure 9). The results show an annual average CH$_4$ emission growth of 2.8±0.6 Tg yr$^{-2}$, a majority of which comes from the energy sector (1.8±0.6 Tg yr$^{-2}$) and wetlands (0.7±0.5 Tg yr$^{-2}$). On a global scale the sector attribution agrees well with INV-FULL but with a smaller magnitude in emission

trends. The reduced growth in INV-FIXED is offset by a higher negative trend in OH concentration (-0.23% yr$^{-1}$), relative to INV_FULL (-0.14% yr$^{-1}$).

In absolute terms OH concentrations are 0.8% lower in INV-FIXED compared to INV-FULL, which acts to offset the lower emissions. OH concentrations for INV-FIXED are 1.8% lower for the 2007-2015 period, relative to the 2003-2006 period, matching the relative change from INV-FULL. Regionally, the largest trends are observed over NA (1.2±0.9 Tg yr$^{-2}$), AM (0.9±0.3 Tg yr$^{-2}$) and AO (0.7±0.4 Tg yr$^{-2}$), with over half of the growth in each of those regions originating from the energy sector. Overall INV-FIXED shows good spatial agreement with INV-FULL when considering sector attribution but the magnitude of emission increases is slightly smaller.

Tropospheric Cl only accounts for a small fraction of the total CH$_4$ sink (~5% or less) (Kirschke *et al*., 2013; Hossaini *et al*., 2016) but, as the kinetic fractionation of Cl reacting with CH$_4$ is more than an order of magnitude greater than that of OH, it is plausible that changes in Cl could contribute to the post-2007 trend in $\delta^{13}CH_4$. Results from an experiment that inverts for CL, INV-CL, (Figure 10) and the sensitivity setup with fixed OH show similar posterior fluxes. This suggests that Cl trends and their effect on $\delta^{13}CH_4$ are unlikely to be an important contributor to the post-2007 CH$_4$ trends, although it is important to note that whilst variability was applied to prior emissions and the OH field, for some years, no variability is applied to the prior Cl field.

### 3.6 Posterior Error

The robustness of the experimental setup is further investigated using the posterior error covariance matrix calculated using equation 5. By splitting the inversion into 12 month intervals emissions later in the year are constrained by fewer observations, possibly only by observations close to the source. The influence of this was investigated and the posterior error was found to be on average 12% higher for December emissions relative to the January emissions, which was broadly consistent between regions and sectors.

Relatively small time independent off-diagonal error correlations are found between different regions and sectors (Figure 11). The posterior covariances produced using equation 5 have been normalised using the corresponding posterior standard deviations to provide posterior correlation values. The largest negative correlation is between EA and NA energy sector emissions, which suggest an artificial trade-off of our results with the increasing NA emissions over 2008-2012 being offset by a decrease in EA emissions over the same period. Overall the results are well constrained by the inversion. Typically, the temporal error correlation is also found to be relatively small, with the exception being the energy sector emissions. Both positive and negative off-diagonal error correlations are found in posterior energy estimates at a monthly resolution, possibly relating to the prior temporal correlation applied, as a result we typically report annual values.

**4 Conclusions**

We have performed a synthesis inversion using a 3-D CTM to investigate the post-2007 renewed growth in atmospheric $CH_4$ and decline in $\delta^{13}CH_4$. This work adds to the results from other studies, which were based on a box-model approach for source and sink attribution based on $CH_4$ and $\delta^{13}CH_4$ observations (e.g. Rigby *et al.*, 2017). By using a 3-D CTM we have been able
to provide detailed monthly regional attribution of 6 different emission sectors and global OH changes, evaluating both the trends over the full 2003-2015 period and shifts that occurred around 2007. We have also been able to validate these results using independent surface sites and recent $XCH_4$ data available from GOSAT and TCCON. The sensitivity of the inversion has been tested for different prior assumptions and uncertainties.

A $CH_4$-only inversion underconstrains the solution with respect to $^{13}CH_4$ observations, resulting in reduced correlation with $\delta^{13}CH_4$ observations (R = 0.60). The agreement of the simulations with observations improved when additional $^{13}CH_4$ observations were used to constrain $CH_4$ fluxes, with the correlation increasing to R = 0.87. The prior model based on published emissions does not capture the $CH_4$ and $\delta^{13}CH_4$ trend both at the assimilated surface site observations and in the non-assimilated GOSAT and TCCON data. In contrast, our derived posterior emission inventories capture both the renewed growth in $CH_4$
and the reduction in $\delta^{13}CH_4$ observed from the assimilated NOAA surface sites from 2007-2015, and compare well with independent surface $CH_4$ and $\delta^{13}CH_4$ observations as well as with GOSAT and TCCON-derived $XCH_4$. The independent validation suggests that, although the $CH_4$ growth rate is better represented in the posterior, it is still underestimated. The posterior model agreement with assimilated surface data and slight bias with validation column data (TCCON and GOSAT) highlights a potential a posteriori model error in total column $CH_4$ concentrations; however, this bias is small. The magnitude
of the contribution of model transport error to this underestimation is unknown. Both prior and posterior simulations underestimate southern hemisphere $CH_4$ concentrations, highlighting possible issues with interhemispheric transport within the model. The lack of independent data around the end of the $CH_4$ 'hiatus' means it is difficult to evaluate model performance over this period (2007).

Our inversion results suggest that the 2007-2015 growth in $CH_4$ can be best explained by a 1.8±0.4% reduction in mean OH, a 12.9±2.7% increase in energy sector emissions, mainly from AM and AO, and a 2.6±1.8% increase in wetland emissions, mainly from EA. The expected increase in atmospheric $\delta^{13}CH_4$ caused by increased energy sector emissions (+0.15‰) is offset mainly by the decrease in OH (-0.12‰), small decrease in biomass burning emissions (-0.08‰) and small increase in wetland emissions (-0.05‰).

When $\delta^{13}CH_4$ is not assimilated the trend in posterior emissions is slightly increased post-2007 and the OH decrease is smaller (-0.3%). By including the $\delta^{13}CH_4$ observations a larger post-2007 OH decrease is required (-1.8%), highlighting the importance of including $\delta^{13}CH_4$ within the inversion.

An alternative scenario, where OH is assumed constant post-2007, requires a -11.5±3.8% decrease in biomass burning emissions, and 13.6±2.7% and 3.6±1.8% increases in energy sector and wetland emissions. These results agree with previous studies, which also assumed constant OH (Nisbet *et al.*, 2016; Schaefer *et al.*, 2016; Worden *et al.*, 2017). Whilst a reduction in OH is found to be, in part, the most likely explanation for the renewed $CH_4$ growth, this alternative scenario with no change in OH provides an alternative explanation for the cause of the post-2007 $CH_4$ growth.

The inversion results suggest Eurasian energy sector emissions are typically overestimated by inventories and previous top-down studies, such as the Global Carbon Budget (Saunois *et al.*, 2016). The reduced EA emissions are found to be offset by an underestimate in all other regions. We find prior annual estimates of biomass burning, waste and rice to be relatively accurate, whilst agricultural estimates are overestimated. Small changes occur in the seasonal cycle of rice emissions and the seasonal range is underestimated in wetland emissions.

Our inversion is found to be robust when small changes are made to uncertainty errors; however, large uncertainty remains around the accuracy of prior emissions. Assuming no prior trend in emissions reduces the required growth rate in both wetland and energy sector emissions, although they remain the main source contribution to the renewed growth post-2007. The reduction in the emission trend is offset by an increased negative trend in OH concentration. Overall the magnitude of the trends inferred varies between experiments but there is consistent agreement that both OH decrease and, wetland and energy sector emission increase contributed to the post-2007 growth.

Our inversion results represent plausible scenarios for variations in $CH_4$ sources and sinks, though several caveats exist. The uncertainties in the sources and sinks are somewhat subjective and we have not considered source signature and kinetic fractionation uncertainty. We have assumed that all uncertainties are independent of each other (excluding energy emissions). We have also not considered variation in other sinks (e.g. $O(^1D)$, soil). The synthesis inversions are performed over coarse spatial regions and only attribute emissions at the monthly scale, future studies should utilise increased observations to provide finer spatial and temporal resolution. The assumption that emissions within a region are correlated limits more specific spatial attribution of sources. Within a region it is likely that some posterior emissions are too high, offset by emissions being too low elsewhere within the domain. The choice of regional aggregation is likely to influence the synthesis inversion, which may result in aggregation errors causing biases in the posterior fluxes (Kaminski *et al.*, 2001). Finally, an important question is whether tropospheric OH has varied in the way suggested by $CH_4$ inversions studies. The processes causing variations in OH are complex and remain poorly quantified. Possible explanations include changes in tropospheric $O_3$ and trends in tropospheric UV radiation related to global stratospheric $O_3$ recovery. If the reduction in available OH due to increased reactive carbon gases is no longer being sufficiently offset by increased emissions of OH-forming nitrogen oxides, then OH concentrations might be in decline (Lelieveld *et al.*, 2004). For example, Itahashi *et al.* (2014) showed a reduction in column $NO_2$ growth

associated with the economic downturn over East Asia between 2008 and 2009, this approximately coincides with the increased $CH_4$ growth.

**Acknowledgments, Samples, and Data**

This work was supported by the NERC MOYA project (NE/N015657/1). MPC and MG acknowledge support from NERC grants GAUGE (NE/K002244/1) and AMAZONICA (NE/F005806/1). RJP was funded via an ESA Living Planet Fellowship with additional funding from the UK National Centre for Earth Observation and the ESA Greenhouse Gas Climate Change Initiative (GHG-CCI). HB was supported by ESA GHG-CCI. CW, RJP and HB acknowledge funding support as part of NERC's National Centre for Earth Observation, contract number PR140015. The TOMCAT runs were performed on the Arc3 supercomputer at U. Leeds. We thank the Japanese Aerospace Exploration Agency, National Institute for Environmental Studies, and the Ministry of Environment for the GOSAT data and their continuous support as part of the Joint Research Agreement. The GOSAT retrievals used the ALICE High Performance Computing Facility at the University of Leicester. NOAA atmospheric $CH_4$ and $\delta^{13}CH_4$ values were obtained from the ESRL GMD Carbon Cycle Cooperative Global Air Sampling Network (esrl.noaa.gov). TCCON atmospheric column $CH_4$ values were obtained from the TCCON data achieve (tccondata.org). The authors would also like to thank Matt Rigby for advice with $^{13}CH_4$ modelling.

All model data used in this study are available through the University of Leeds ftp server. For access please contact M.Chipperfield@leeds.ac.uk.

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

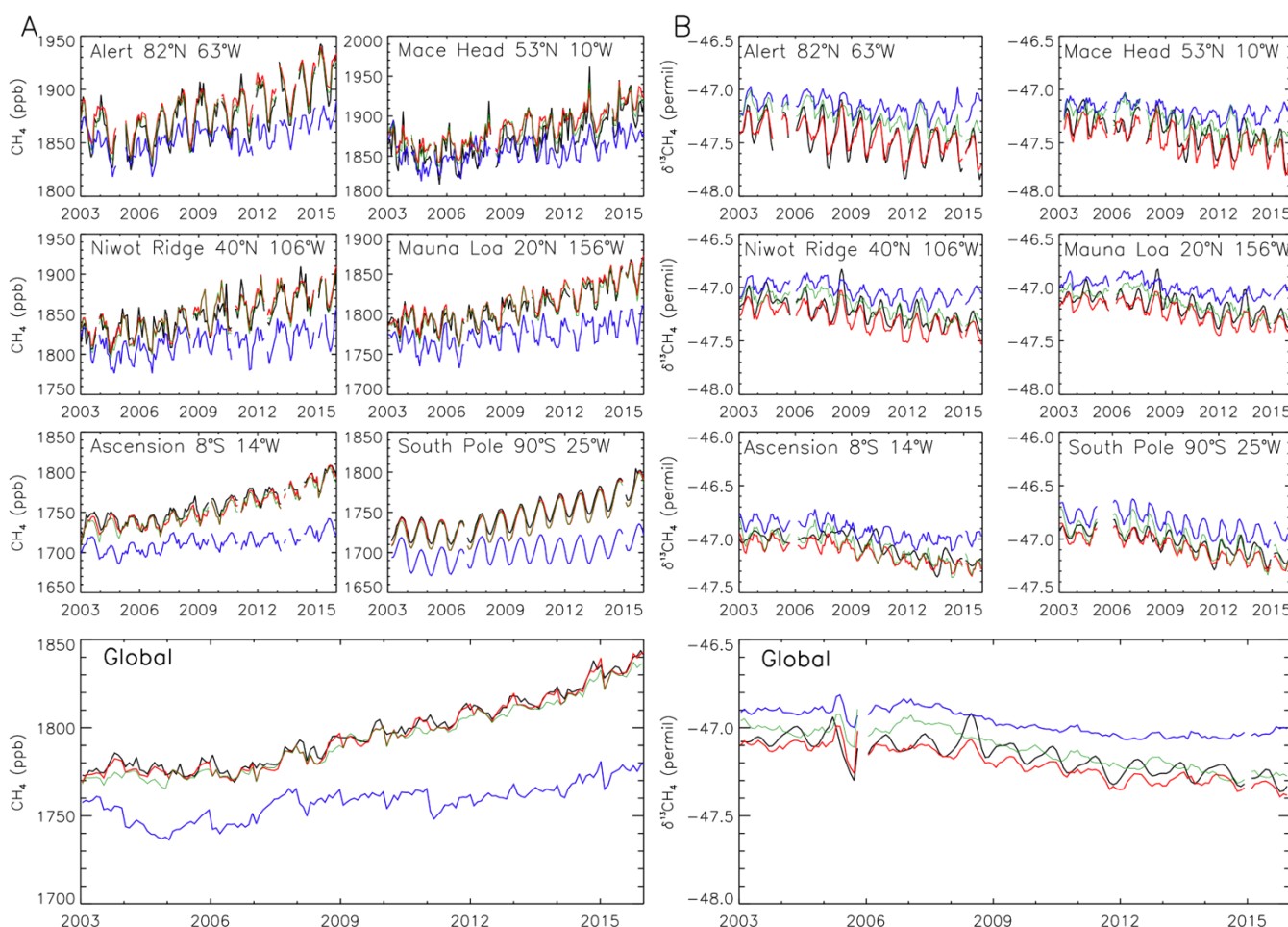

**Figure 1. (a) Observed surface CH₄ (ppb, black line) from 2003 to 2015 at 6 selected NOAA sites and global mean. Also shown are results from TOMCAT simulations using prior emission estimates (blue line), posterior estimates based on a CH₄ synthesis inversion (INV-CH4, green line) and posterior estimates based on a combined CH₄ and δ¹³CH₄ synthesis inversion (INV-FULL, red line). (b) Same as (a) but for observed and modelled δ¹³CH₄. Global averages are based on site interpolations onto 180 1°-latitude bins, which are weighted by surface area.**

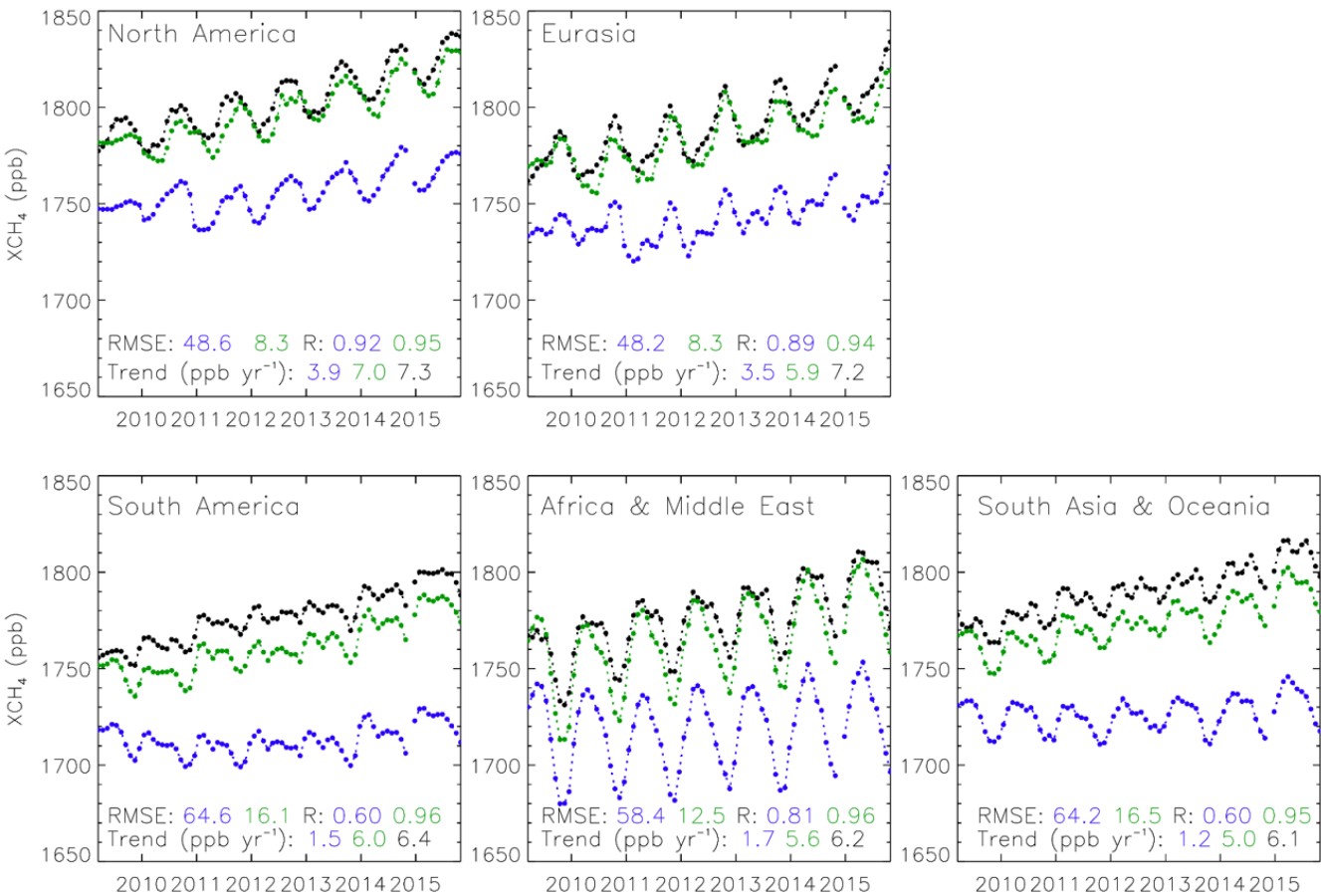

**Figure 2. Monthly mean XCH₄ volume mixing ratio (ppb) from GOSAT between April 2009 and December 2015 (black line) for 5 emission regions. Also shown are results from TOMCAT simulations with prior (blue) and posterior (green) emission estimates, both with GOSAT averaging kernels applied. Correlation coefficients, RMSE and growth rates of the model simulations and GOSAT in each region are shown in the panels.**

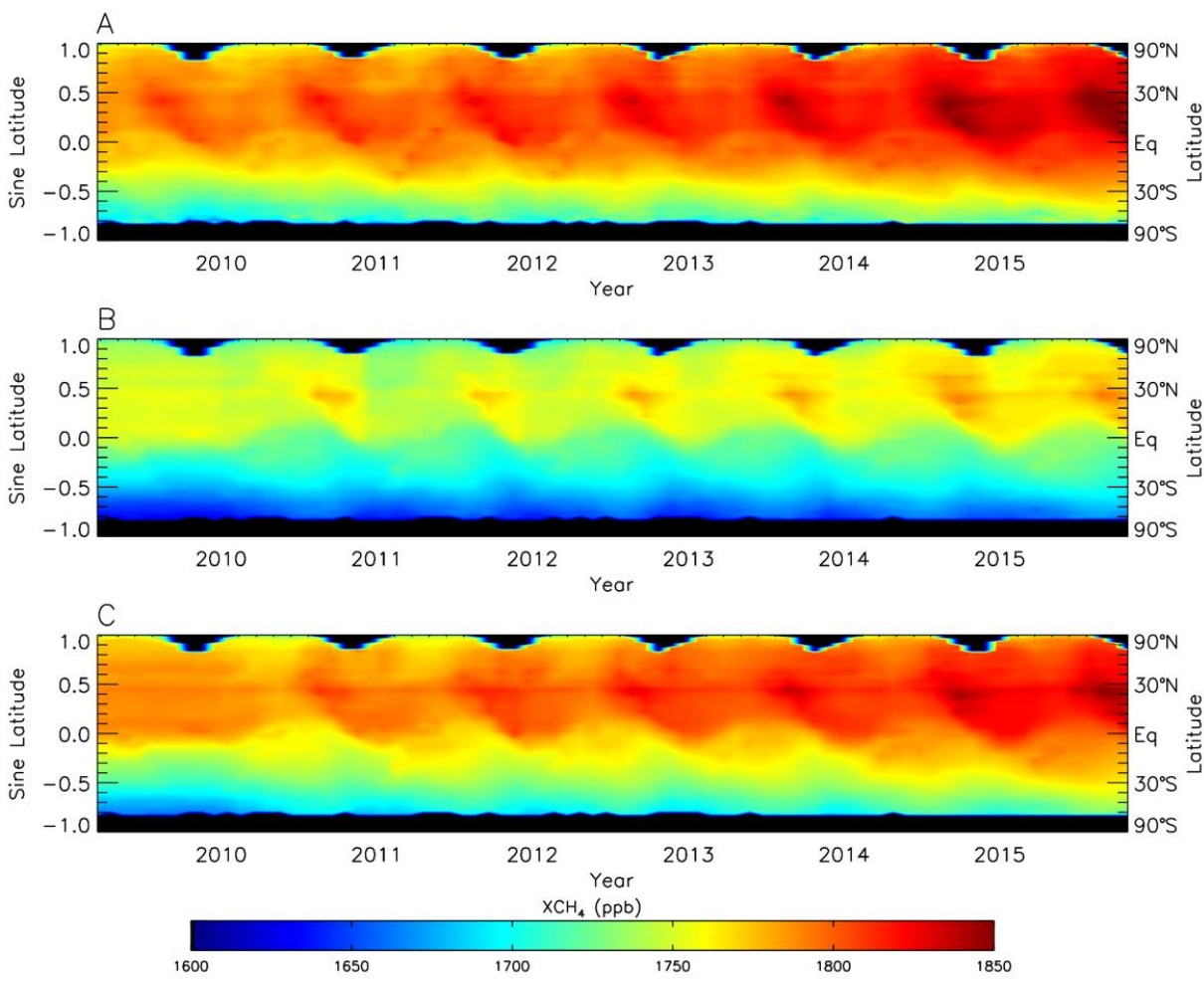

**Figure 3. (a) Zonally averaged monthly mean XCH₄ volume mixing ratio (ppb) from GOSAT between April 2009 and December 2015 plotted against the sine of latitude, where black denotes missing values. (b and c) Same as (a) but for TOMCAT simulations with prior and posterior emission estimates, respectively. GOSAT averaging kernels are applied to model simulations.**

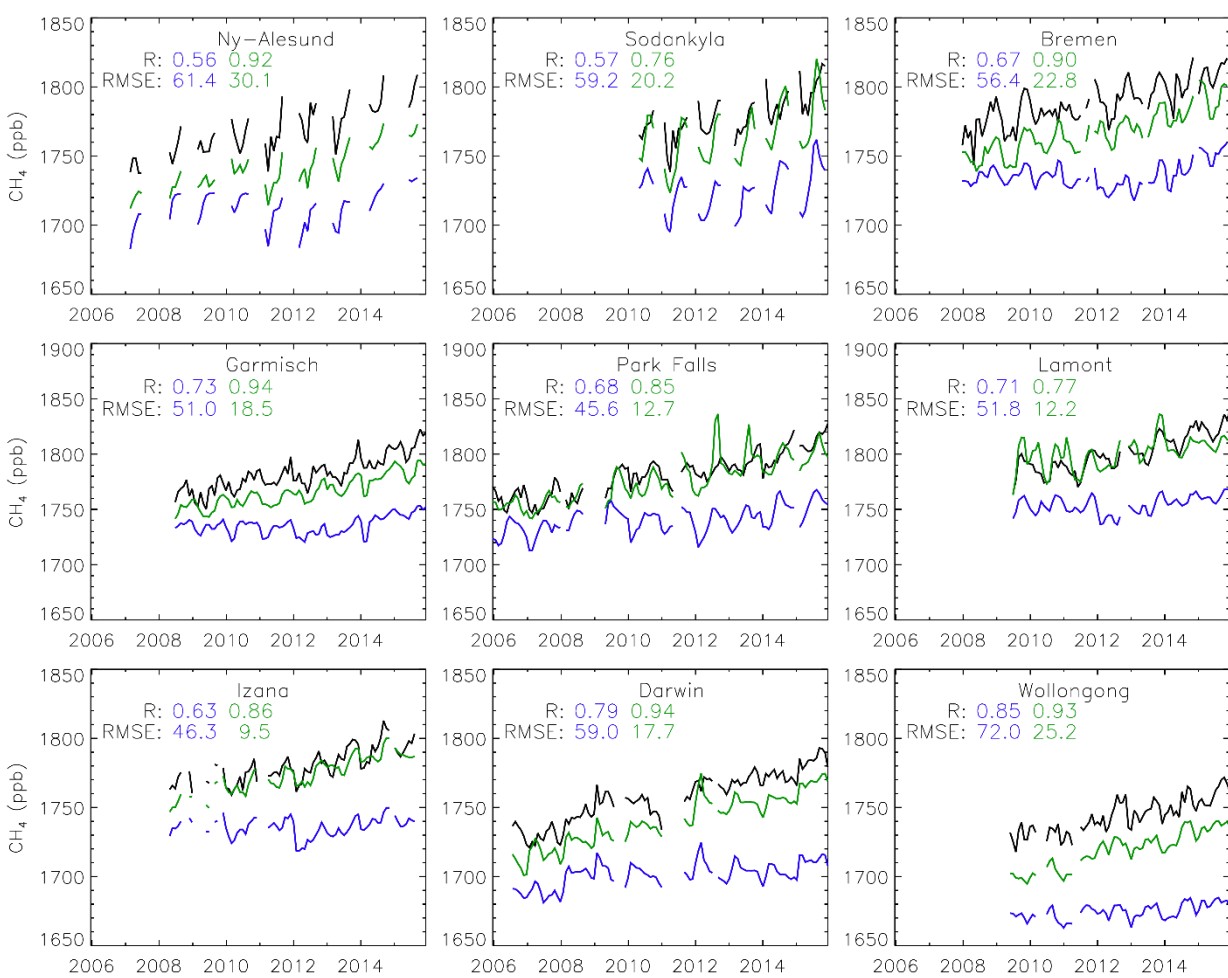

**Figure 4.** Observed monthly mean XCH₄ volume mixing ratio (ppb) (blackline) at 9 TCCON sites. Also shown are results from TOMCAT simulations with prior (blue) and posterior (green) emission estimates, both with TCCON averaging kernels applied. Correlation coefficients and RMSE of the model simulations compared with TCCON are shown for each site.

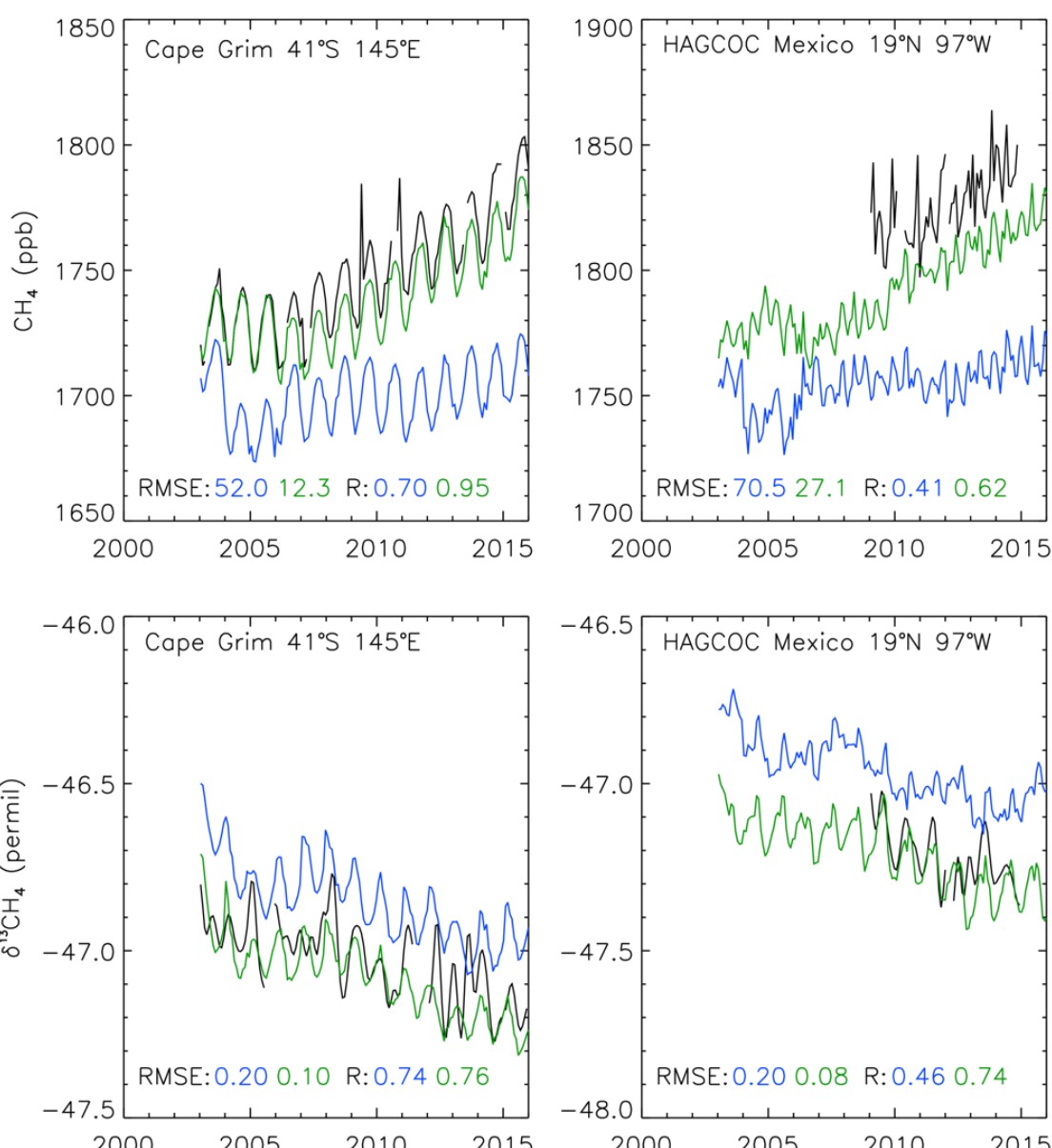

**Figure 5. Observed surface CH4 (top) and δ¹³CH4 (bottom) from 2003 to 2015 at 2 independent NOAA sites (black line). Also shown are results from TOMCAT simulations using prior emission estimates (blue line), and posterior estimates based on a combined CH4 and δ¹³CH4 synthesis inversion (INV-FULL, green line). RMSE and correlation coefficients of the model simulations compared with observations are shown for each site.**

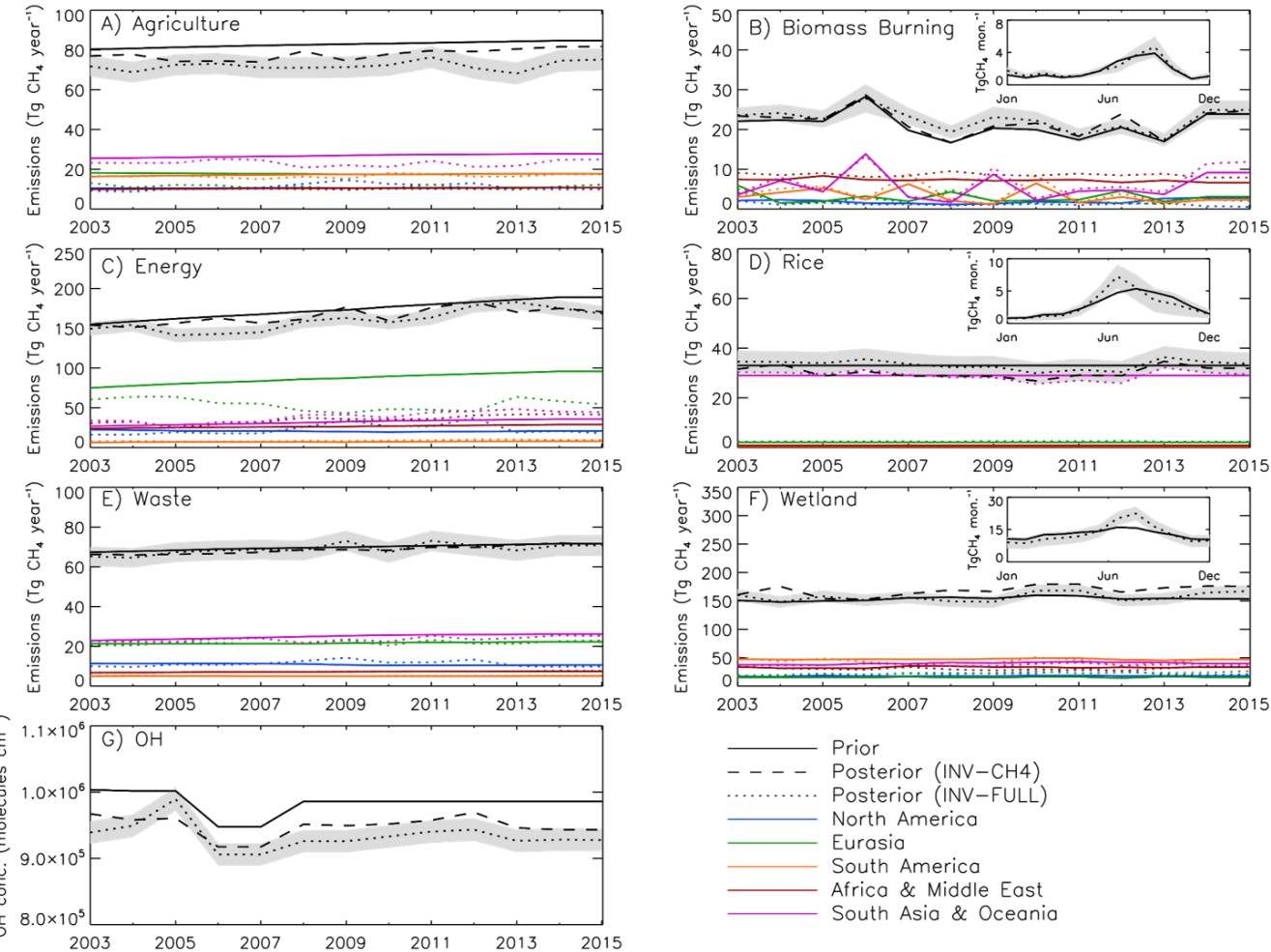

**Figure 6. (a-f) Annual CH₄ emissions (Tg CH₄ year⁻¹) from different sectors for global prior (black solid line), INV-CH4 (black dashed line) and INV-FULL posterior (black dotted line) estimates. Regional estimates are also displayed for North America (blue), Eurasia (green), South America (orange), Africa and Middle East (red) and South Asia and Oceania (purple). (g) Prior and posterior global OH estimates for the same period. Shaded region denotes posterior error *A* for INV-FULL (see Eq. (5) in text).**

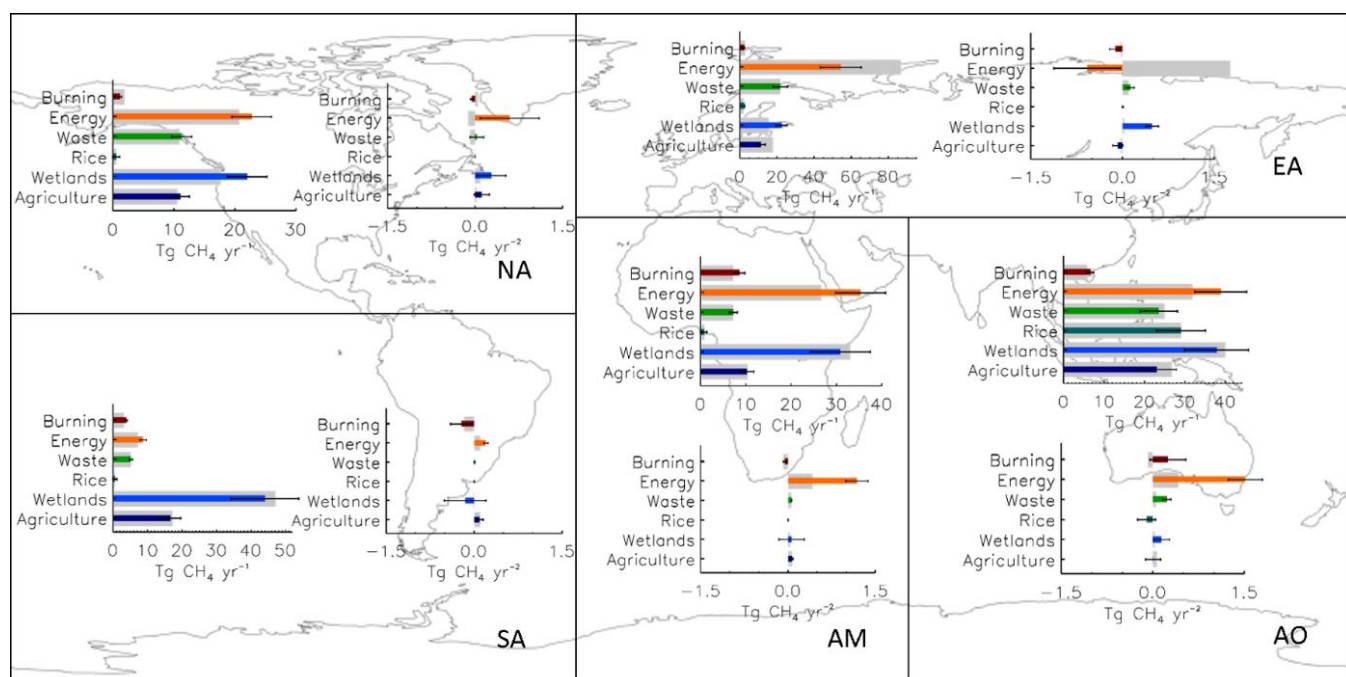

**Figure 7. Map showing regional annual mean CH₄ emissions (Tg CH₄ year⁻¹) and yearly change in emissions (Tg CH₄ year⁻²) calculated as a linear regression between 2003 and 2015 for INV-FULL (thin coloured bars) and prior (thick grey bars) estimates. Error bars represent one standard deviation of the mean posterior emissions and posterior regression errors. Note that the black borders indicate the 5 regions used for the flux partitioning.**

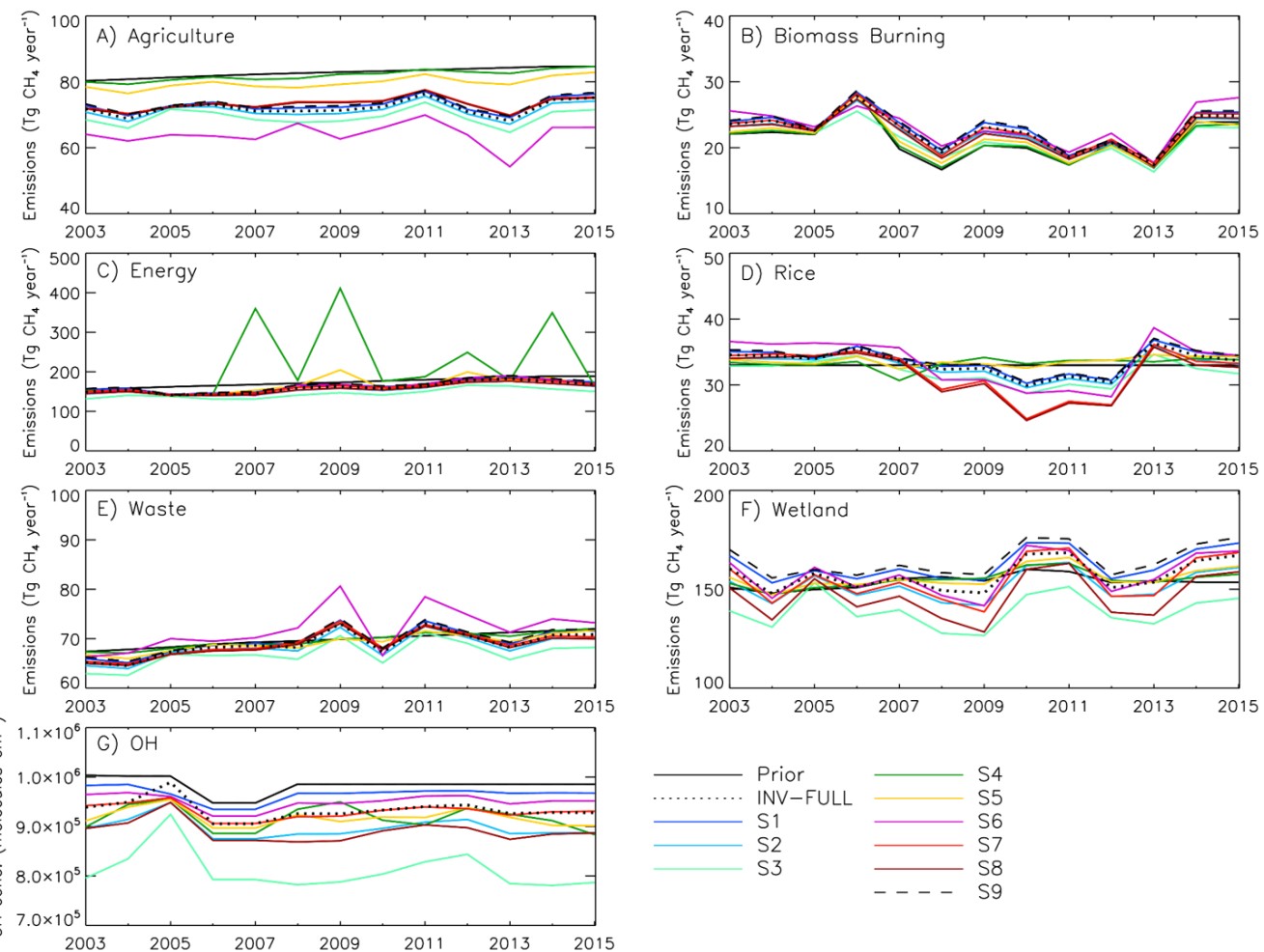

**Figure 8.** (a-f) Annual mean CH₄ emissions (Tg CH₄ year⁻¹) from different sectors for global prior (black solid line) and INV-FULL (black dotted line) estimates. (g) Same as (a-f) but for global mean OH (molecules cm⁻³). Additional lines in each panel show sensitivity inversions with different emission and OH uncertainties (coloured lines), and an inversion assuming no change in OH (black dashed line).

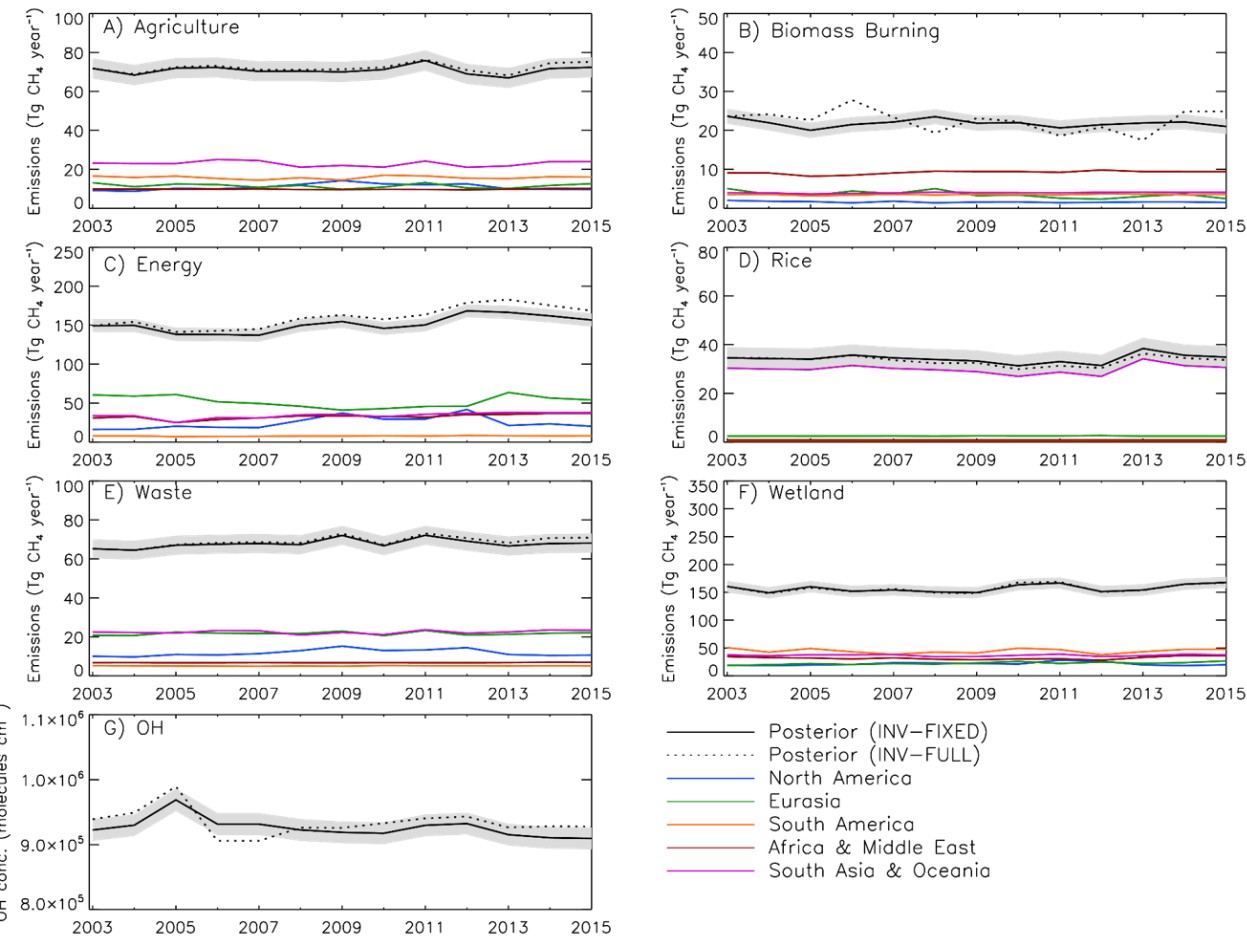

**Figure 9. (a-f)** Annual CH₄ emissions (Tg CH₄ year⁻¹) from different sectors for INV-FIXED (black solid line) and INV-FULL posterior (black dotted line) estimates. Regional estimates are also displayed for North America (blue), Eurasia (green), South America (orange), Africa and Middle East (red) and South Asia and Oceania (purple). **(g)** Prior and posterior global OH estimates (molecules cm⁻³) for the same period. Shaded region denotes posterior error $A$ (see Eq. (5) in text).

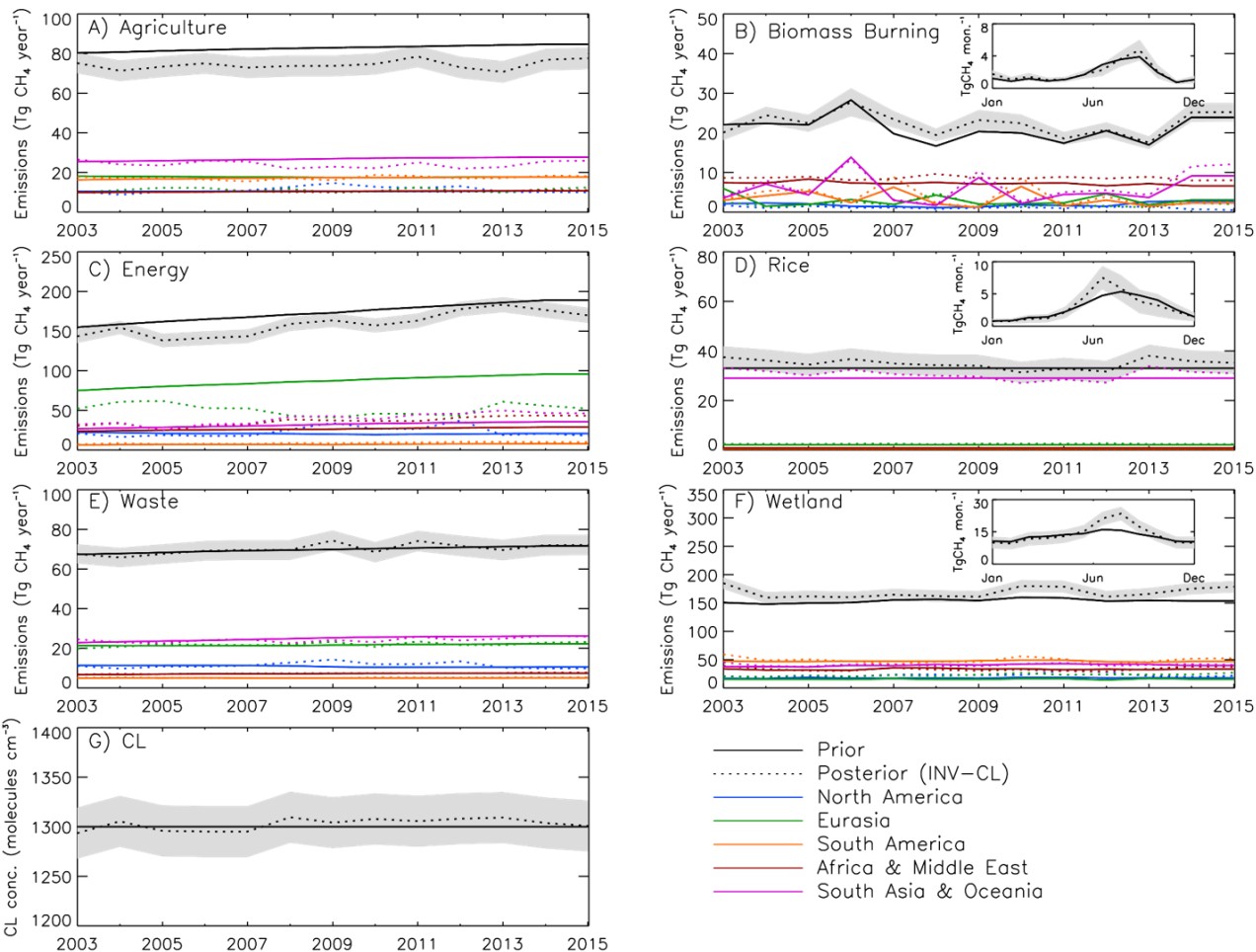

**Figure 10. (a-f) Annual CH₄ emissions (Tg CH₄ year⁻¹) from different sectors for global prior (black solid line) and INV-CL (black dotted line) estimates. Regional estimates are also displayed for North America (blue), Eurasia (green), South America (orange), Africa and Middle East (red) and South Asia and Oceania (purple). (g) Prior and posterior global tropospheric Cl estimates for the same period. Shaded region denotes posterior error A (see Eq. (5) in text).**

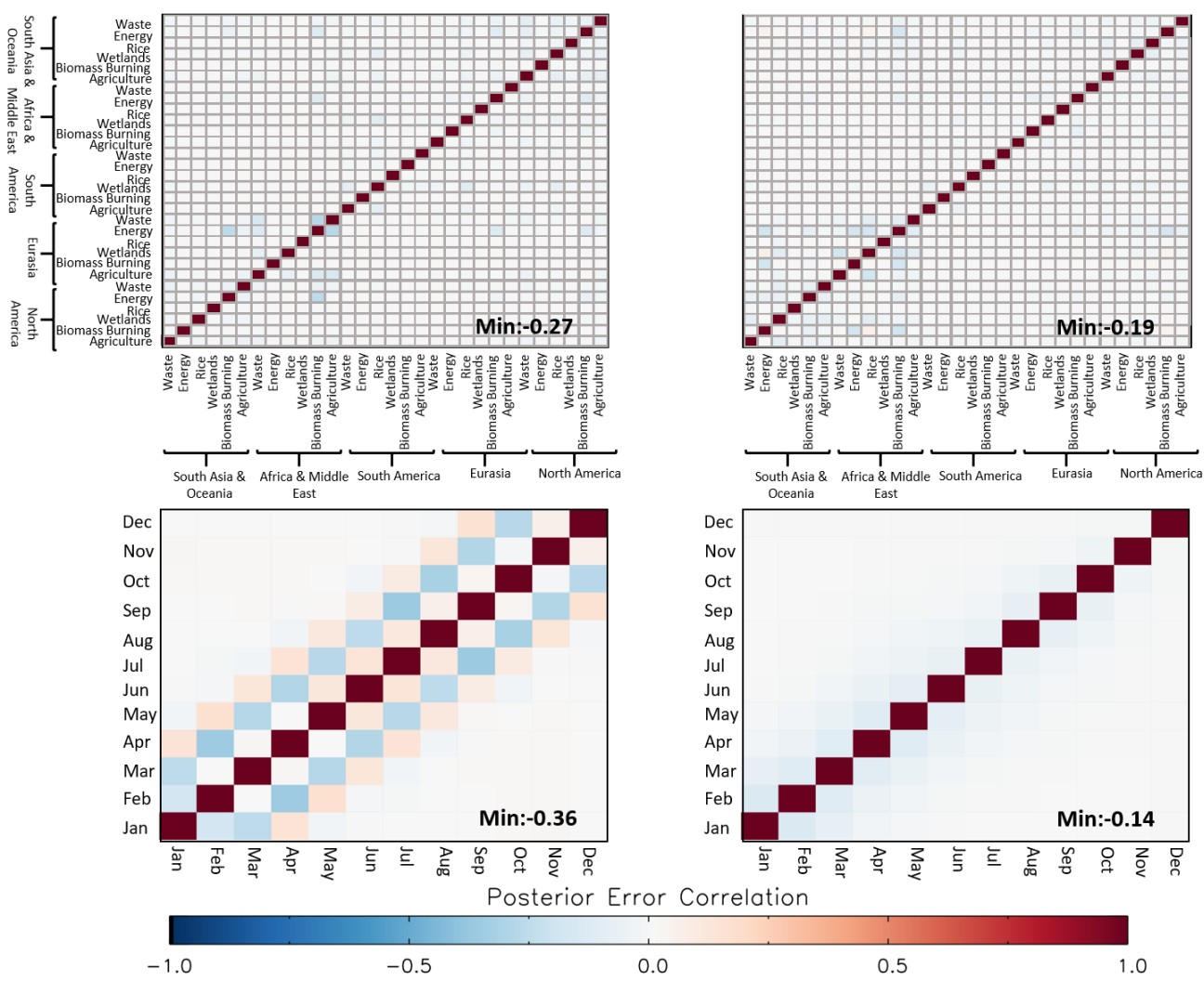

**Figure 11. Posterior error correlation matrix for all regions and sectors for January (top-left) and July (top-right) 2015. Posterior error correlation matrix for 12 months of 2015 for the Eurasian energy sector (bottom-left) and South American wetlands (bottom-right). The plots show a subset of the total posterior error correlation matrix as an example. Values are calculated by normalising the posterior covariances using the corresponding posterior standard deviations.**

| Source | Interannual variability | $\delta^{13}CH_4$ (‰) | Emissions for 2003-2015 (Tg CH$_4$ yr$^{-1}$) | |
|---|---|---|---|---|
| | | | Prior | Posterior |
| Agriculture (excluding rice) | Yes | -61.3 | 82.8 | 72.1±5.4 |
| Biomass Burning | Yes | -22.2 | 21.1 | 22.5±2.2 |
| Energy | Yes | -42.6 | 173.5 | 160.1±8.9 |
| Rice | No | -62.0 | 33.0 | 33.3±4.4 |
| Waste | Yes | -55.6 | 69.8 | 68.9±5.1 |
| Wetlands | Yes | -61.0 | 153.8 | 157.2±10.2 |
| Soil Methantrophy (negative emission) | No | 22.0 | -27.2 | - |
| **Sink** | | Kinetic isotope effect ($^{12}CH_4/^{13}CH_4$) | Average concentration for 2003-2015 (molecules cm$^{-3}$) | |
| | | | Prior | Posterior |
| OH | Yes (2003-2007) | 1.0039 | $0.98\times10^6$ | $0.93\times10^6$ |
| Cl | No | 1.06 | $1.3\times10^3$ | $1.3\times10^3$ |
| O($^1$D) | No | 1.013 | - | - |

**Table 1. Source and sink isotope signatures used in the TOMCAT 3-D CTM. Values for prior emissions (Kirschke *et al.*, 2013 McNorton *et al.*, 2016a, Schwietzke *et al.*, 2016) and isotope signatures (Saueressig *et al*., 2001; Mikaloff-Fletcher *et al.*, 2004; Feilberg *et al.*, 2005; Whiticar *et al.*, 2007; Schwietzke *et al.*, 2016) are based on previous studies. Note that the soil sink is not optimised and modelled as a negative emission. Posterior emission estimates are shown with posterior error estimates.**

| Site Code | Site Name | Latitude (°N) | Longitude (°E) | Altitude (m) | Measurements |
|---|---|---|---|---|---|
| ALT | Alert, Canada | 82.5 | -62.5 | 190.0 | $CH_4$, $\delta^{13}CH_4$ |
| ASC | Ascension Island, UK | -8.0 | -14.4 | 85.0 | $CH_4$, $\delta^{13}CH_4$ |
| AZR | Terceira Island, Portugal | 38.8 | -27.4 | 19.0 | $CH_4$, $\delta^{13}CH_4$ |
| BRW | Barrow, USA | 71.3 | -156.6 | 11.0 | $CH_4$, $\delta^{13}CH_4$ |
| CBA | Cold Bay, USA | 55.2 | -162.7 | 21.3 | $CH_4$ |
| HBA | Halley Station, UK | -75.6 | -26.2 | 30.0 | $CH_4$ |
| ICE | Storhofdi, Iceland | 63.4 | -20.3 | 118.0 | $CH_4$ |
| KUM | Cape Kumukahi, USA | 19.5 | -154.8 | 3.0 | $CH_4$, $\delta^{13}CH_4$ |
| MHD | Mace Head, Ireland | 53.3 | -9.9 | 5.0 | $CH_4$, $\delta^{13}CH_4$ |
| MLO | Mauna Loa, USA | 19.5 | -155.6 | 3397.0 | $CH_4$, $\delta^{13}CH_4$ |
| NWR | Niwot Ridge, USA | 40.1 | -105.6 | 3523.0 | $CH_4$, $\delta^{13}CH_4$ |
| PAL | Pallas-Sammaltunturi, Finland | 68.0 | 24.1 | 565.0 | $CH_4$ |
| PSA | Palmer Station, USA | -64.9 | -64.0 | 10.0 | $CH_4$ |
| RPB | Ragged Point, Barbados | 13.2 | -59.4 | 15.0 | $CH_4$ |
| SMO | Tutuila, American Samoa | -14.2 | -170.6 | 42.0 | $CH_4$, $\delta^{13}CH_4$ |
| SPO | South Pole, USA | -90.0 | -24.8 | 2810.0 | $CH_4$, $\delta^{13}CH_4$ |
| STM | Ocean Station M, Norway | 66.0 | 2.0 | 0.0 | $CH_4$ |
| SUM | Summit, Greenland | 72.6 | -38.4 | 3209.5 | $CH_4$ |
| THD | Trinidad Head, USA | 41.1 | -124.2 | 107.0 | $CH_4$ |
| WLG | Mt. Waliguan, China | 36.3 | 100.9 | 3810.0 | $CH_4$, $\delta^{13}CH_4$ |
| ZEP | Ny-Alesund, Norway/Sweden | 78.9 | 11.9 | 474.0 | $CH_4$ |

Table 2. NOAA measurements from 2003 to 2015 used in the synthesis inversions of $CH_4$ (Dlugokencky *et al*., 2017) and $\delta^{13}CH_4$ (White *et al*., 2017).

| Site Name | Latitude (∘N) | Longitude (∘E) | Altitude (km) | Reference |
|---|---|---|---|---|
| Ny-Alesund, Norway | 78.9 | 11.9 | 0.02 | Notholt *et al*. 2017a |
| Sodankyla, Finland | 67.4 | 26.6 | 0.19 | Kivi *et al*. 2017 |
| Bremen, Germany | 53.1 | 8.9 | 0.03 | Notholt *et al*. 2017b |
| Garmisch, Germany | 47.5 | 11.1 | 0.74 | Sussmann *et al*. 2017 |
| Park Falls, USA | 45.9 | -90.3 | 0.44 | Wennberg *et al*. 2017a |
| Lamont, USA | 36.6 | -97.5 | 0.32 | Wennberg *et al*. 2017b |
| Izana, Spain | 28.3 | -16.5 | 2.37 | Blumenstock *et al*. 2017 |
| Darwin, Australia | -12.5 | 130.9 | 0.04 | Griffith *et al*. 2017a |
| Wollongong, Australia | -34.4 | 150.9 | 0.03 | Griffith *et al*. 2017b |

**Table 3. TCCON sites (Wunch *et al*. 2011) used for evaluation of the TOMCAT simulations.**

| Region | Prior Annual Emissions by Sector (Tg CH$_4$ year$^{-1}$) | | | | | | |
|---|---|---|---|---|---|---|---|
| | Biomass Burning | Energy | Waste | Rice | Wetlands | Agriculture | Total |
| N. America | 1.9 | 20.7 | 10.9 | 0.5 | 17.7 | 10.6 | 71.3 |
| Eurasia | 2.9 | 87.0 | 21.7 | 2.1 | 15.6 | 17.8 | 165.9 |
| S. America | 3.2 | 7.2 | 5.1 | 0.5 | 47.2 | 17.2 | 82.3 |
| Africa & Middle E. | 7.2 | 26.6 | 7.2 | 0.8 | 33.2 | 10.4 | 85.4 |
| S. Asia & Oceania | 5.8 | 32.0 | 24.9 | 29.0 | 40.1 | 26.8 | 177.8 |
| Global | 21.1 | 173.5 | 69.8 | 33.0 | 153.8 | 82.8 | 571.0 |
| Region | Posterior Annual Emissions by Sector (Tg CH$_4$ year$^{-1}$) | | | | | | |
| | Biomass Burning | Energy | Waste | Rice | Wetlands | Agriculture | Total |
| N. America | 1.2±0.6 | 22.8±4.0 | 11.2±2.4 | 0.5±0.1 | 22.0±4.0 | 11.0±2.2 | 68.7±7.3 |
| Eurasia | 2.3±1.0 | 54.4±11.8 | 21.7±4.8 | 2.5±0.6 | 22.5±3.7 | 11.2±3.1 | 114.6±13.9 |
| S. America | 3.8±1.1 | 8.7±1.7 | 5.1±1.2 | 0.5±0.1 | 44.0±10.6 | 16.6±3.8 | 78.7±15.1 |
| Africa & Middle E. | 8.6±1.9 | 35.4±6.3 | 7.2±1.6 | 0.8±0.2 | 30.8±7.4 | 10.2±2.3 | 93.1±12.5 |
| S. Asia & Oceania | 6.6±1.6 | 38.9±7.2 | 23.6±5.4 | 29.0±6.7 | 37.9±8.6 | 23.1±5.6 | 159.1±16.7 |
| Global | 22.5±2.9 | 160.1±15.8 | 68.9±7.8 | 33.3±6.8 | 157.2±16.5 | 72.1±8.1 | 537.5±26.5 |

Table 4. Regional CH$_4$ emissions based on prior (top) and synthesis inversion estimates (bottom) between 2003 and 2015. Note the total global emission, but not the total regional emissions, include the supplementary emissions (geological, hydrates, oceans and termites). Uncertainties are also shown for posterior emissions, all prior emissions have a 50% uncertainty.

| Region | Annual Emission Growth by Sector (Tg CH$_4$ year$^{-2}$) | | | | | | |
|---|---|---|---|---|---|---|---|
| | Biomass Burning | Energy | Waste | Rice | Wetlands | Agriculture | Total |
| N. America | -0.06 | +0.59 | +0.03 | +0.00 | +0.28 | +0.11 | +0.95 |
| Eurasia | -0.12 | -0.58 | +0.13 | +0.00 | +0.48 | -0.08 | -0.17 |
| S. America | -0.22 | +0.20 | +0.01 | +0.00 | -0.15 | +0.09 | -0.06 |
| Africa & Middle E. | -0.05 | +1.18 | +0.06 | +0.00 | +0.06 | +0.07 | +1.33 |
| S. Asia & Oceania | +0.25 | +1.51 | +0.23 | -0.10 | +0.14 | +0.00 | +2.03 |
| Global | -0.20 | +2.91 | +0.46 | -0.10 | +0.81 | +0.20 | +4.08 |

Table 5. Regional CH$_4$ emission growth trends based on synthesis inversion estimates between 2003 and 2015.

| Simulation | Annual Emission by Sector for the 2003-2006 Period (Tg CH$_4$ year$^{-1}$) | | | | | | |
|---|---|---|---|---|---|---|---|
| | **Biomass Burning** | **Energy** | **Waste** | **Rice** | **Wetlands** | **Agriculture** | **Total** |
| INV_FULL | 24.5 | 146.9 | 66.3 | 34.6 | 154.4 | 71.6 | 518.9 |
| INV_CH4 | 24.4 | 156.0 | 66.3 | 31.1 | 160.7 | 75.9 | 529.8 |
| INV_FIXED | 21.8 | 143.8 | 66.1 | 34.6 | 155.4 | 71.1 | 514.6 |
| | Annual Emission by Sector for the 2007-2015 Period (Tg CH$_4$ year$^{-1}$) | | | | | | |
| | **Biomass Burning** | **Energy** | **Waste** | **Rice** | **Wetlands** | **Agriculture** | **Total** |
| INV_FULL | 21.6 | 165.9 | 70.1 | 32.7 | 158.4 | 72.3 | 545.8 |
| INV_CH4 | 20.9 | 169.9 | 69.6 | 30.0 | 171.9 | 78.7 | 557.7 |
| INV_FIXED | 21.8 | 154.5 | 68.7 | 34.0 | 158.1 | 70.9 | 536.1 |
| | Difference in Annual Emission Between 2007-2015 and 2003-2006 (Tg CH$_4$ year$^{-1}$) | | | | | | |
| | **Biomass Burning** | **Energy** | **Waste** | **Rice** | **Wetlands** | **Agriculture** | **Total** |
| INV_FULL | -2.9 | +19.0 | +3.8 | -1.9 | +4.0 | +0.7 | +26.9 |
| INV_CH4 | -3.5 | +13.9 | +3.3 | -1.1 | +11.2 | +2.8 | +27.9 |
| INV_FIXED | 0.0 | +10.7 | +2.6 | -0.6 | +2.7 | -0.2 | +21.5 |

**Table 6. Posterior annual CH$_4$ emission for the period of near-zero atmospheric growth (2003-2006) and the renewed growth (2007-2015) based on three different inversion simulations. Note the total emissions, include the supplementary emissions (geological, hydrates, oceans and termites).**

| Source/sink | Sensitivity Test Error | | | | | | | | | |
|---|---|---|---|---|---|---|---|---|---|---|
| | Control | S1 | S2 | S3 | S4 | S5 | S6 | S7 | S8 | S9 |
| **Wetlands** | 50% | 50% | 50% | 50% | 10% | 20% | 100% | 100% | 100% | 50% |
| **Rice** | 50% | 50% | 50% | 50% | 10% | 20% | 100% | 100% | 100% | 50% |
| **Agriculture (excluding rice)** | 50% | 50% | 50% | 50% | 10% | 20% | 100% | 50% | 50% | 50% |
| **Waste** | 50% | 50% | 50% | 50% | 10% | 20% | 100% | 50% | 50% | 50% |
| **Energy** | 50% | 50% | 50% | 50% | 10% | 20% | 100% | 50% | 50% | 50% |
| **Biomass Burning** | 50% | 50% | 50% | 50% | 10% | 20% | 100% | 50% | 50% | 50% |
| **OH** | 2% | 1% | 3% | 10% | 2% | 2% | 2% | 2% | 3% | 0% |

**Table 7. Suite of inversion sensitivity experiments with varying errors on source and sink estimates.**

| Simulation | Annual Emission by Sector for the 2003-2006 Period (Tg CH$_4$ year$^{-1}$) | | | | | | |
|---|---|---|---|---|---|---|---|
| | Biomass Burning | Energy | Waste | Rice | Wetlands | Agriculture | Total |
| Control | 24.5 | 146.9 | 66.3 | 34.6 | 154.4 | 71.6 | 518.9 |
| S1 | 24.9 | 150.4 | 66.8 | 35.0 | 158.7 | 72.3 | 530.4 |
| S2 | 24.1 | 143.5 | 65.9 | 34.3 | 150.2 | 70.9 | 507.3 |
| S3 | 23.2 | 135.2 | 64.7 | 33.4 | 139.9 | 69.2 | 479.6 |
| S4 | 23.7 | 146.7 | 67.8 | 33.3 | 150.7 | 80.3 | 515.2 |
| S5 | 23.9 | 146.5 | 67.3 | 33.7 | 152.1 | 78.5 | 515.9 |
| S6 | 25.0 | 146.6 | 68.2 | 36.3 | 155.2 | 63.4 | 524.3 |
| S7 | 24.7 | 147.7 | 66.2 | 34.7 | 152.1 | 72.1 | 518.5 |
| S8 | 24.2 | 143.8 | 66.0 | 34.4 | 145.4 | 71.9 | 504.9 |
| S9 | 25.1 | 152.0 | 67.0 | 35.2 | 160.7 | 72.6 | 535.8 |
| | Annual Emission by Sector for the 2007-2015 Period (Tg CH$_4$ year$^{-1}$) | | | | | | |
| | Biomass Burning | Energy | Waste | Rice | Wetlands | Agriculture | Total |
| Control | 21.6 | 165.9 | 70.1 | 32.7 | 158.4 | 72.3 | 545.8 |
| S1 | 22.0 | 170.5 | 70.7 | 33.2 | 164.0 | 73.2 | 560.6 |
| S2 | 21.2 | 161.3 | 69.4 | 32.3 | 152.8 | 71.5 | 530.9 |
| S3 | 20.2 | 149.7 | 67.8 | 31.3 | 138.8 | 69.3 | 493.7 |
| S4 | 19.9 | 250.3 | 70.5 | 33.3 | 156.9 | 82.7 | 626.5 |
| S5 | 20.4 | 174.0 | 70.1 | 33.6 | 157.7 | 80.3 | 550.7 |
| S6 | 22.5 | 169.8 | 73.5 | 32.4 | 158.8 | 64.3 | 560.2 |
| S7 | 21.7 | 167.0 | 70.3 | 30.7 | 156.1 | 74.0 | 545.4 |
| S8 | 21.1 | 161.7 | 69.9 | 30.3 | 147.1 | 73.8 | 527.4 |
| S9 | 22.2 | 172.6 | 71.0 | 33.3 | 166.5 | 73.6 | 567.2 |

Table 8. Posterior annual CH$_4$ emission for the period of near-zero atmospheric growth (2003-2006) and the renewed growth (2007-2015) based on suite of inversion sensitivity experiments with varying errors on source and sink estimates. Note the total emissions, include the supplementary emissions (geological, hydrates, oceans and termites).