# Peer review of "Attribution of recent increases in atmospheric methane through 3-D inverse modelling"

_Atmospheric Chemistry and Physics, 2018_

## Referee Comment (RC1) · Anonymous Referee #2 · 18 Jul 2018

General comments :

This paper by McNorton et al. addresses the important question of attributing the unexplained recent changes in atmospheric methane concentrations (since 2007) to methane sources and sinks. The validation part is good and useful.

The main originality of the paper is to include 13CH4 observations and OH changes in an inversion based on a 3D transport model, as compared to previous crude box modelling approaches (including Nature papers !)

The results of this paper are very important for the methane & climate communities, although, in the present form of the paper, they are not valorized because of : - a rather confusing organization of section 3 and conclusions. - a general lack of details

and precisions in all sections - a lack of precise comparison with previous analyses

My main demands are (see specific comments) - to rewrite sections 3.2 and 3.3 clearly presenting and separating results & analysis of o mean emission and sinks versus their changes, o results & analysis global scale versus regional o results & analysis of emissions versus the sinks - to comment on all emissions (anthropogenic microbial emission poorly commented) - to report emission changes in Tg/yr between 2 time periods and not in trends (Tg/yr2), - to add a table with emissions changes, - to add a discussion section where comparison with other studies can be grouped

In short, this is an important paper to be published in ACP, material is mostly there but a profound rewriting/organization of the results & conclusions sections is needed.

Specific comments :

P2-L15-20: please note that these mean isotopic signatures are associated with rather large range. It may be worth writing also that total source signature is -51/-53‰

P2 L25 : "although they emphasised that the problem is not very well constrained by existing data Âż I suggest to be more precise : although these two studies cannot discard the hypothesis that OH is not changing.

P3 L5 : please define shortly here "synthesis inversion" (3D modelling, reduction of the size of flux and observation spaces to solve the inverse problem, ...) focusing on the imorvement compared to box models.

P3 L18 : what is a one-year inversion spin-up ? please detail a bit.

P3 L21-22 : what is the influence of this choice ? what do you take for geological emissions ? It might be worth making a sensitivity test by taking the values from Saunois et al (2016) (update of the Kirschke paper, please quote) instead of the Schwietzke paper.

P3 L26 : I would be worth mentioning which of your sources is prescribed in the prior

with internannual variability. Maybe in table 1.

P3 L30 : I understand that you compute monthly response functions using the forward model? Please specify this here.

P4 L 5 : why increasing OH and Cl ? please justify this choice ?

P4 L9-10 and P5 L9-14: not clear. How do you deal with the long-term equilibration of 13CH4 (e.g. Tans 97 paper) with 1-yr inversions ?

P4 L11 : Âń For the inversion including OH concentrations" : this suggests that there are inversions without OH in the state vector. Please clarify.

P4 : I understand that isotopic signature are not optimized in this procedure. Please precise this point.

P5 L4-7 : Bousquet et al 2011 addressed this point and tested a second iteration with only small impact on the inversion results, so consistent with your hypothesis. It might be worth quoting.

P5 L9 : "The model OH is constrained by CH4 and $\delta$13CH4" : this is a weak constraint as many combination of total source and mean OH can fit the atmospheric changes. Please notice it here ? With such a configuration you largely depend on the prior for the mean emissions and sinks so I would not insist a lot in the paper on the posterior versus prior comparison but more on budget changes with time and between your different inversions.

P6 L10-11 : putting only one value of uncertainty for all stations is a bit crude as model error will not be the same for remote sites of the southern hemisphere and continental sites of the northern hemisphere. More refinement is needed here or at least a sensitivity test varyng observational errors

P6 : It might worth doing a sensitivity test with more atmospheric observations, when appearing in the network. The apparition of stations is an issue but can help analyzing

regional gradients more safely. As you perform yearly inversions, why not adding each year the stations appearing in your inversions ?

P7 L22 : "slow inter-hemispheric transport within the model" : please provide the Inter Hemispheric Time and/or a reference for this possibly from transcom experiments ?

P7 L27 : For Garmisch did you try to extract the station at different level in your model ?

P7 L29 : 21.4 ppb is still a quite large value. Can you at least make hypotheses to explain them ?

P8 – sect3.2 : "OH concentrations in INV-FULL and INV-CH4 are relatively constant throughout the period 2007-2015 (Figure 5) but these values are smaller by 1.8±0.4% and 0.3±0.5%" : unclear : do OH is constant or diminishing. Please clarify. Also, I find a bit strange to start by the section by the sink and not by source changes.

P8 L9-10 : it may be good to refer to the sensitivity test on OH (S9) here.

P8 l11-15 : mixing the mean changes compared to the prior and the time changes from 2003-2006 to post-2007 period is confusing. What about change in agriculture flux ? I suggest to group discussions on the mean sources and sinks global and regional (table 4) and then address the changes (table 5)..

P8 L24 : how did you estimate the 30% for OH and 60%/10% values ? Did you use S9? Please justify.

P8 : The choice to report changes in trends (Tg yr-2) is a bit technical. A suggestion would be to report emission change between two periods (e.g. 2003-2005 and 2012-2015) in Tg/yr and quantify the % of contributions from this.

Table 5 : there are some values worth to comment in your analysis : increase emissions from NA ? dipole +0.59 / -0.58 for energy between NA and EA ? wetland increase in Eurasia ? Why ? visible in other studies ?

Section 3.2 : You do not comment waste sector (+0.46 in your table 5). Indeed, anthropogenic microbial emission contribute almost as much as wetlands (0.46 and 0.2 trends globally versus 0.8 for wetlands). Please add comments on anthropogenic microbial emission changes.

P9 L5-9 : you have very few stations constraining NA inland emissions. It should be notices here as the increased inferred emissions over NA is a hot topic. Again including NA inland stations in a sensitivity inversion seems necessary to confirm such a result. In any case this has to be commented here.

P10 : It is a bit difficult to follow all the trends provided and to compare them to the standard inversion. I suggest to make a table with results of sensitivity test for global scale compared to INV-FULL(in Tg/yr difference between 2 periods and not trends in Tg/yr2). Then you can more clearly comment on the differences in the main text.

P10 L7 : "the magnitude in post-2006 changes is typically increased in S9", please add something like : which is normal considering that constant OH as compared to decreasing OH in INV-FULL requires more emission change to match atmospheric observations

P10 L21-22 : Again, pleas acknowledge here that global total OH versus total emissions are not very well constrained without an external proxy as many commbiantion can match the growth rate. You largely rely to the prior in this case so I would not insist a lot on the posterior versus prior comparison but more on budget changes with time and between your different inversions.

P11 L15-30 : This comparison with EDGAR should be in in a discussion section between 3.3 and conclusions where you could compare your results with other studies. More references to previous results would be good e.g. Pouter et al. 2017 for wetlands, Saunois et al., 2017 ACP for all sources, more precise comparison about OH with Rigby and turner papers. . .

P11 L21-22 : "As a result emissions from these regions are influenced by posterior emission changes and assumed to be underestimated in both magnitude and growth rate in the prior" : unclear to me, please rephrase.

P11 L27 I do not see this -2.2 Tg/yr2 in table 5 ? please clarify.

P12 L11 : you do not believe your results ? this sentence introduce confusion. Please rephrase it or remove it.

P12 L13 : Saunois paper is not an inventory by a synthesis of inventories and inversions. Please rephrase.

P12 L27 : Limitation of synthesis inversions (monthly means, coarse regions. . .) should also be mentioned here.

P12 L31 : what about NO2 decrease in Asia in the late 2000s ? Please mention this hypothesis as well.
* * *

---

## Referee Comment (RC2) · Anonymous Referee #1 · 9 Sep 2018

The manuscript "Attribution of recent increases in atmospheric methane through 3-D inverse modelling" by J. McNorton and co-workers adds to the considerable number of studies focussing on the question why atmospheric methane concentrations started rising again post 2007 after a decade of virtually stable concentrations. Although many suggestions based on different methodologies have been brought forward in recent years, there is still no final consensus on which source or atmospheric process or which combination of these has caused the renewed rise. By applying a 3D transport model to methane concentration and to 13CH4/12CH4 ratios the authors add an interesting and valuable study to the discussion that is in general worth publishing in ACP. However, I have several comments concerning the applied inversion method and model validation that need to be addressed before publication. I would also like to suggest that

the authors check the manuscript carefully again for a more precise language. Some examples are given below, but there are many more where a more precise wording would improve the readability of the text.

Major comments

1) Regional sub-division

What is the rational for the regional division applied in the transport model and the inversion? Especially the large EA and AO emission regions combining countries and regions with very different socio-economic developments in the last decades are very questionable choices. As the inversion is set up right now it can, for example, not distinguish between Western Europe with well established and generally decreasing CH4 emissions from most sectors from emissions in Russia or North-East Asia. These are areas with potentially growing emissions from different sectors in the last decades. Although, these trends may be presented in the a priori it seems more likely that there are uncorrelated uncertainties in the emission estimates for these areas. Similar arguments can be found for a required sub-division between south-east Asia and Australia. In the end, the current sub-division alters the derived regional trends in the a posteriori emission very questionable. For example opposing regional errors in the a priori trends in these large regions may alter it impossible for the inversion to correctly correct these trends. Instead the missing/excessive emissions may be lain down in/removed from regions for which little direct constraint is available from the utilised set of observations, but only a more global sensitivity exists in the model (such as the AM region). Maybe not surprisingly these are the regions for which the authors find the strongest changes in a posteriori emissions, a result that somewhat differs from conclusions in previous work. The regional sub-division certainly needs some further justification. This could be done by a more in-depth validation of the model performance at surface sites in contrasting areas like Europe vs. East Asia. For this the use of additional surface observations should be considered (see major comment 3).

2) General applicability of inversion method

As stated correctly on page 2 line 23, Rigby et al. (2017) and Turner et al. (2017) both conclude that the problem of the post 2007 methane rise may be under-constrained using the observed CH4 concentrations, 13CH4/12CH4 ratios and other tracers. Their conclusion is based on simpler box-model simulations without detailed regional division of CH4 emissions. In the present study an even larger number of unknowns is optimised through the inversion. Wouldn't this mean that the individual elements of the state vector are even less well constraint? The authors should spend some time justifying why their more detailed results should be better constrained than those from box-model analyses. In this context it may be worth looking at the covariances in the a posteriori emission and OH factor as well. Large negative covariances may indicate that the inversion cannot clearly distinguish between regions and sectors.

3) Surface observations

The authors base their inverse flux estimates on a limited set of surface observations (22 flask sampling sites). This may be justified in order to keep the influence of CH4 concentrations to 13CH4/12CH4 ratios similar, with the latter only being available at this limited number of locations. However, for validation purposes there would be many more CH4 observations available worldwide (flask and continuous). These should be evaluated as independent observations as well and may better than GOSAT and TCON observations demonstrate the success of the inverse flux estimate.

Minor comments

P2L26f: Although the global a priori emissions by source category are available in Table 1 and regionally divided a posteriori emissions are given in Table 4, I am missing the same kind of information for the a priori. An additional table in the style of Table 4 but for the a priori emissions should be added.

P4L7ff: If I correctly understand the inversion setup, the inversion step is performed on

batches of 12 months. Does this mean that the emissions from the previous year are not influenced by the observations of the next year at all? Meaning that January observations will not influence December emissions from the previous year? This would result in December, but probably also November and October, emissions always being constraint by less observations than emissions in other months and, therefore, probably are less corrected from their a priori values and/or show systematically larger a posteriori uncertainties than emissions in other months. Was this observed in the a posteriori factors?

P5L16: The wording is not very precise here. J is a cost function and the inversion will find its minimum. J is not a minimisation function. Instead equation 4 represents the analytical minimum of equation 3.

P5L24: R is not the covariance matrix of the observations alone. R contains the observation/model mismatch covariance. Later this fact is taken care of by adding a model uncertainty to R, but it should be correctly introduced here.

P6L6: Was any month-to-month variability of the emissions included in the a priori? If yes where was it taken from?

P6L10f: This is a bit simplistic since the model uncertainty most likely varies with the location of the observation and the question how representative the model grid cell can be for a given site. There have been many different approaches in the past on how to assign site-dependent model uncertainties and, hence, this point should be justified a bit more.

P7L6 and elsewhere: A lot of this RMSE is due to a bias in the a priori simulation. It would be better to calculate a bias-corrected RMSE instead. The bias could be mentioned separately. In general it would be nice to include all these comparison statistics in a table as well (in the main text for all discussed inversions and observational data sets and in the supplementary material for all sensitivity inversions).

P7L23f: I don't think it is the model that is growing here. What about 'simulated atmospheric methane growth rates' instead?

P7L27f: This behaviour is very strange. For all other sites an increase in concentrations from a priori to a posteriori simulations was observed. Why not for Garmisch, a central European site not too far away from the Bremen site, where differences in the a priori and a posteriori simulations are as expected? One potential source of mismatch may be the location of Garmisch at the northern edge of the Alps, potentially introducing large mismatches due to smoothed model topography. Still this would not explain the lack of an increase from a priori to a posteriori. Although a detail, this needs to be checked again.

P8L5 and Figure 5: The estimated a posteriori OH time series should also be compared with work by other authors (e.g. Rigby et al. 2017). If OH is really the main driver of the post 2007 CH4 rise it would be good to know how TOMCAT OH compares to previous work.

P8L15: A reference to Table 6 should be added here.

P9L11: A reference to Figure 5 should be added here.

P9L28f: How similar? These numbers are not given anywhere. One can only guess them from the figures. A table (like Table 4) with the a posteriori emissions for the INV-CL case should be provided and the same for all sensitivity inversion (supplement).

P9L28f: How is the a posteriori performance for this experiment (S4)? Just because one sensitivity run gives different a posteriori emissions it doesn't have to be wrong. But if it also fails to reproduce the observations, then the given conclusion may be correct.

P10L31f: This sounds a bit like the authors of Rigby et al. worked on the current study as well. Which is not the case. This work may extend the previous work by using a more complex transport model, but other than that the approaches are fairly different

and unrelated (inversion system, used observations, etc.). So I would not write that it extends the work of Rigby or others, but rather it adds to the results gained by others.

P11L7: 'larger errors'. What kind of errors? Needs to be repeated here.

P11L7f: The sentence 'The constraint improves when the $\delta$13CH4 observations are introduced' should be re-written to be more precise. What about: 'The agreement of the simulations with observations improved when additional $\delta$13CH4 observations are used to constrain CH4 fluxes.'.

P11L12: This conclusion is just based on the different trend compared with GOSAT, whereas the trend in surface observations was captured well in the a posteriori simulation. Does that mean that there is a potential trend in the bias between GOSAT and surface observations? Would there be any GOSAT validation studies that may provide some clarification?

P11L15f: Once again: There are more surface observations available than used in this study. They should be used for validation during this critical period.

P11L29: It is unclear which period is referred to here? Table 5 suggests a growth rate in the energy sector of the AO region of 1.5 Tg yr-2 the text states -2.2 Tg yr-2. What is correct?

P12, 1st paragraph: This section should also repeat what was stated in the introduction concerning previous inverse modelling studies (P2L21ff), especially since the presented results contradict/correct these earlier findings.

Figure1: Is is impossible to see the red dotted lines in many of the sub-panels (also the ones for $\delta$13CH4). Either the figure needs to be enlarged/split or an additional color and solid line should be used for INV-CH4.

Table1, Table4, Table6: These should also contain the uncertainty estimates.

Table1: Maybe I missed this before, but does the missing number for the soil sink mean

that it was neglected completely? If it was only not-optimised its value should still be part of this table.

---

## Author Response (AR1)

**Response to reviewers' comments**

We would like to thank the reviewers for their helpful comments. These are repeated below (in italics) followed by our responses.

**Reviewer 1**

My main demands are (see specific comments) - to rewrite sections 3.2 and 3.3 clearly presenting and separating results & analysis of o mean emission and sinks versus their changes, o results & analysis global scale versus regional o results & analysis of emissions versus the sinks - to comment on all emissions (anthropogenic microbial emission poorly commented) - to report emission changes in Tg/yr between 2 time periods and not in trends (Tg/yr2), - to add a table with emissions changes, - to add a discussion section where comparison with other studies can be grouped.

We agree that both section 3.2 and 3.3 could be made clearer by providing more details and separating the results as follows:

- Prior and posterior comparison.
- Posterior trends both globally and regionally.
- Source and sink attribution from inversion.
- Integrate sensitivity.

We agree that the reviewer suggestions will improve the manuscript significantly and thank him/her for them. We have addressed the following specific comments relating to the general remarks above.

*P2 L15-20: Please note that these mean isotopic signatures are associated with rather large range. It may be worth writing also that total source signature is -51/-53‰.*

We have included the total source signature as suggested and highlight the categories given are in a broad range.

*P2 L25: "although they emphasised that the problem is not very well constrained by existing data I suggest to be more precise: although these two studies cannot discard the hypothesis that OH is not changing.*

We agree that the discussion of the conclusions drawn by Rigby et al., 2017 and Turner et al., 2017 was not detailed enough and as a result we have now also commented that their results could not discard the hypothesis of no OH change.

P3 L5: please define shortly here "synthesis inversion" (3D modelling, reduction of the size of flux and observation spaces to solve the inverse problem, ...) focusing on the improvement compared to box models.

We agree that giving a short sentence describing the synthesis inversion and comparing it to box models would be useful here. This has been included.

P3 L18: what is a one-year inversion spin-up? please detail a bit.

We have appended the sentence to explain the one year spin-up is used to optimise the model CH4 and  $\delta^{13}$ CH4 concentration fields relative to the observations.

P3 L21-22: what is the influence of this choice? what do you take for geological emissions ? It might be worth making a sensitivity test by taking the values from Saunois et al (2016) (update of the Kirschke paper, please quote) instead of the Schwietzke paper.

We agree that the relevance to the Kirschke study is outdated and we have updated the reference to the more recent Saunois et al. study. The decision to scale to Schwietzke et al. estimates was based

on their development of isotope source signatures, which we felt was relevant for this paper. However, we agree that the Saunois et al. study provides more thorough estimates, and a sensitivity study using both estimates would be interesting, a comparison of the budgets between the two studies is however beyond the scope of this work.

P3 L26: I would be worth mentioning which of your sources is prescribed in the prior with interannual variability. Maybe in Table 1.

OK. We have updated Table 1 to include which source/sinks vary interannually.

P3 L30: I understand that you compute monthly response functions using the forward model? Please specify this here.

Yes. We have updated the text to specify the monthly emissions can be used to assess variability.

**P4 L5: Why increasing OH and Cl? Please justify this choice?**

By adjusting the OH, the sensitivity can be diagnosed, and this sensitivity remains the same whether fields are increased or decreased. A small feedback is present in the model setup due to  $CH_4$  loss rate being dependent on  $CH_4$  concentration. To reduce the impact of this, the sensitivity simulations only adjust the OH concentrations by a small amount (1%). This has been added to the text.

**P4 L9-10 and P5 L9-14: Not clear. How do you deal with the long-term equilibration of 13CH4 (e.g. Tans 97 paper) with 1-yr inversions?**

The inversions are performed for monthly emissions, although the total inversion length (13 years) comprises a series of the 1-year inversions mentioned. We have included that the 1-year inversions are performed for computational reasons, and by rerunning the forward model with posterior fluxes we are able to provide initialisation fields for the subsequent year. In effect, this serves as a single 13-year inversion for the purpose of long-term equilibration, because the previous year posterior fluxes influence future concentration fields. The posterior fluxes are not influenced by observations beyond the 1 month window; however the timescale of changes in the isotopic signature are still captured by the inversion.

Our results suggest that given the source signature change in the total posterior emissions from 2007, the response time of atmospheric  $\delta^{13}$ CH4 is comparable to that of CH4 at the spatial scales resolved in this study. This suggests the long-term equilibration of  $\delta^{13}$ CH4 shown by Tans (1997) using a box model approach is not applicable to the 3D CTM used here.

**P4: I understand that isotopic signature are not optimized in this procedure. Please precise this point.**

We agree that more details were needed. In the submitted paper there was a comment P12 L30 that mentions this, but we have added a new sentence on P4 to highlight this point.

**P5 L4-7: Bousquet et al 2011 addressed this point and tested a second iteration with only small impact on the inversion results, so consistent with your hypothesis. It might be worth quoting.**

**We thank the reviewer for pointing this out and have updated the text to include the finding of Bousquet *et al.* (2011).**

P5 L9: "The model OH is constrained by CH4 and  $\delta$ 13CH4": this is a weak constraint as many combination of total source and mean OH can fit the atmospheric changes. Please notice it here? With such a configuration you largely depend on the prior for the mean emissions and sinks so I would not insist a lot in the paper on the posterior versus prior comparison but more on budget changes with time and between your different inversions.

We agree that multiple plausible emission/sink scenarios could exist to fit the observations, we highlight that a paucity of observations prevent a single solution. However, the 3D model approach,

over the box-model approach reduces the combination spread. We have updated the results and summary to remove the focus on prior v posterior and focus more on budget changes as suggested (Tables 4, 5 and 6).

*P6* L10-11: Putting only one value of uncertainty for all stations is a bit crude as model error will not be the same for remote sites of the southern hemisphere and continental sites of the northern hemisphere. More refinement is needed here or at least a sensitivity test varying observational errors.

We agree that the magnitude of the model transport error will differ between sites, however the quantification of this transport error is beyond the scope of this work. We have added text outlining that the assumption is made as an estimate and not quantitatively derived.

*P6: It might worth doing a sensitivity test with more atmospheric observations, when appearing in the network. The apparition of stations is an issue but can help analysing regional gradients more safely. As you perform yearly inversions, why not adding each year the stations appearing in your inversions?*

We agree both approaches could be adopted. If we included additional sites as and when they became available, then we would gain more information from the inversion. We did not adopt this approach for the same reason as we did not use GOSAT retrievals in the inversion, intermittently adding observations would influence trend results and could result in disjointed posterior estimates, which result from the inclusion of new observations. Therefore, for long-term trend detection we opted not to append the observation set. For more accurate instantaneous posterior estimates we agree new observations should have been used, but as the key aim of the paper was to investigate the trend before and after 2007 we chose to omit new observations.

*P7 L22: "slow inter-hemispheric transport within the model": please provide the Inter Hemispheric Time and/or a reference for this possibly from Transcom experiments?*

We have added text referring to the Patra et al. (2011) Transcom study with reference to the Inter Hemispheric transport.

**P7 L27: For Garmisch did you try to extract the station at different level in your model?**

The averaging kernel is applied from the nearest model pressure level to the TCCON surface pressure and upwards, which is done to remove lower levels from the model output. It is possible that sub-grid scale variations in concentration due to orography, which are not accounted for due to smoothing in the coarse resolution model, is a cause of the model bias. We have not attempted to extract station information at different model levels due to the complex gradients related to orography and vertical profiles.

**P7 L29: 21.4 ppb is still a quite large value. Can you at least make hypotheses to explain them?**

We agree these are still relatively large errors, although they are more than halved relative to the prior and are approximately 1% of the concentration. A possible reason for this is that only surface observations are used in the inversion and not column information measured by TCCON. We have added text outlining this limitation.

P8 - sect3.2: "OH concentrations in INV-FULL and INV-CH4 are relatively constant throughout the period 2007-2015 (Figure 5) but these values are smaller by  $1.8\pm0.4\%$  and  $0.3\pm0.5\%$ ": unclear : do OH is constant or diminishing. Please clarify. Also, I find a bit strange to start by the section by the sink and not by source changes.

We agree it was somewhat unclear and have restructured the sentence to clarify that OH concentrations between 2007 and 2015 are constant but are lower relative to the previous years (2003-2006). In regard to structuring, the section has now been reformulated following the suggestions within this comment and the following comments.

**P8 L9-10: It may be good to refer to the sensitivity test on OH (S9) here.**

We agree the original structure could be difficult to follow. We have now referenced the sensitivity section here for completeness.

*P8* L11-15: Mixing the mean changes compared to the prior and the time changes from 2003-2006 to post-2007 period is confusing. What about change in agriculture flux? I suggest to group discussions on the mean sources and sinks global and regional (table 4) and then address the changes (table 5).

We have rewritten this section to clearly separate out mean attribution and trends in sources and sinks.

**P8 L24: How did you estimate the 30% for OH and 60%/10% values? Did you use S9? Please justify.**

We agree this was not made clear in the results, and we have updated the text to explain. The inversion results were used in a simple box model to attribute contributions of each source and sink to the observed CH4 trend. The details of the box model are found in McNorton et al (2016b).

*P8:* The choice to report changes in trends (Tg yr-2) is a bit technical. A suggestion would be to report emission change between two periods (e.g. 2003-2005 and 2012-2015) in Tg/yr and quantify the % of contributions from this.

We agree that reporting the results as the shift between the two periods is clearer. As the reviewer mentioned the structure of the section did not flow and this was one of the reasons. As a result, we have restructured the section to bring these two sets of analysis into the same part. We have kept the more technical growth rates (Tg yr-2), as they provide information about the rate of change, but have also now included Tg/yr and the % contribution.

Table 5: There are some values worth to comment in your analysis: increase emissions from NA? dipole +0.59 / -0.58 for energy between NA and EA? wetland increase in Eurasia? Why? visible in other studies?

The inversion ability to constrain NA and EA energy sector emissions independently reveals potential issues as discussed in Section 3.2. As a result, we have added in two sentence to describe a potential limitation in the model inversion system when used to distinguish between NA and EA energy sector emissions. The spatial distribution in wetland trends from previous studies remains uncertain and as a result is not used to inform the results of this study.

Section 3.2: You do not comment waste sector (+0.46 in your table 5). Indeed, anthropogenic microbial emission contribute almost as much as wetlands (0.46 and 0.2 trends globally versus 0.8 for wetlands). Please add comments on anthropogenic microbial emission changes.

We agree that whilst the magnitude of the waste emission changes is not as large as either energy sector or wetland changes, the relative change is larger and therefore should be commented on. We have now included this in our results.

*P9 L5-9: You have very few stations constraining NA inland emissions. It should be notices here as the increased inferred emissions over NA is a hot topic. Again including NA inland stations in a sensitivity inversion seems necessary to confirm such a result. In any case this has to be commented here.*

We agree the sparse observations over NA and EA may be the cause of some of these anomalous results, we have added this in as a caveat to the results.

P10: It is a bit difficult to follow all the trends provided and to compare them to the standard inversion. I suggest to make a table with results of sensitivity test for global scale compared to INV-FULL (in Tg/yr difference between 2 periods and not trends in Tg/yr2). Then you can more clearly comment on the differences in the main text.

We agree and have added this table in as Table 8, the results discussion now reflects what is shown in the table for clarity.

P10 L7: "the magnitude in post-2006 changes is typically increased in S9", please add something like: which is normal considering that constant OH as compared to decreasing OH in INV-FULL requires more emission change to match atmospheric observations.

We agree that this point should be made clear, i.e. that the results are expected due to the removal of the OH sensitivity. We have added a line explaining this.

P10 L21-22: Again, please acknowledge here that global total OH versus total emissions are not very well constrained without an external proxy as many combination can match the growth rate. You largely rely to the prior in this case so I would not insist a lot on the posterior versus prior comparison but more on budget changes with time and between your different inversions.

We agree that the paper needed to focus more on the trend/shift over the period and less on the comparison with the prior. We have added a comment here and elsewhere in the paper to emphasise this point.

P11 L15-30: This comparison with EDGAR should be in a discussion section between 3.3 and conclusions where you could compare your results with other studies. More references to previous results would be good e.g. Pouter et al. 2017 for wetlands, Saunois et al., 2017 ACP for all sources, more precise comparison about OH with Rigby and Turner papers.

We agree that the ordering of discussion is confusing and have changed it as suggested. We have included reference to Saunois et al., (2017). Poulter et al. is not included because it only covers a subset of the period studied. We have also given quantitative comparisons to the Rigby and Turner papers.

P11 L21-22: "As a result emissions from these regions are influenced by posterior emission changes and assumed to be underestimated in both magnitude and growth rate in the prior": unclear to me, please rephrase.

We agree this sentence is not clear. We have modified it to describe that the inversion system can attribute fluxes and trends at a regional level, but cannot diagnose spatial patterns at a sub-regional scale (for example national).

P11 L27: I do not see this -2.2 Tg/yr2 in Table 5? Please clarify.

We agree this was not clear. We have clarified that the EDGAR comparison is for 2003-2012, when EDGAR data are available, while the figure in Table 5 (-0.58 Tg/yr2) is for 2003-2015. The EA energy sector emissions appeared to rebound slightly in later years.

P12 L11: You do not believe your results? This sentence introduce confusion. Please rephrase it or remove it.

OK. We have modified the sentence to state that whilst our findings provide the most likely explanation for the cause of the renewed growth, an alternative scenario could exist whereby OH remains unchanged; however, this is considered less likely.

P12 L13: Saunois paper is not an inventory by a synthesis of inventories and inversions. Please rephrase.

We agree we did not describe the Saunois paper correctly. We have modified this to explain the combination of inventories and top-down studies.

P12 L27: Limitation of synthesis inversions (monthly means, coarse regions...) should also be mentioned here.

We agree this is a caveat to the study and have added this in.

P12 L31: What about NO2 decrease in Asia in the late 2000s? Please mention this hypothesis as well.

We have included a reference to the decreased  $NO_2$  growth rate in the late 2000s, which approximately coincides with the renewed  $CH_4$  growth.

**Reviewer 2**

What is the rational for the regional division applied in the transport model and the inversion? Especially the large EA and AO emission regions combining countries and regions with very different socio-economic developments in the last decades are very questionable choices. As the inversion is set up right now it can, for example, not distinguish between Western Europe with well-established and generally decreasing CH4 emissions from most sectors from emissions in Russia or North-East Asia. These are areas with potentially growing emissions from different sectors in the last decades. Although, these trends may be presented in the a priori it seems more likely that there are uncorrelated uncertainties in the emission estimates for these areas. Similar arguments can be found for a required sub-division between south-east Asia and Australia. In the end, the current sub-division alters the derived regional trends in the a posteriori emission very questionable. For example, opposing regional errors in the a priori trends in these large regions may alter it impossible for the inversion to correctly correct these trends. Instead the missing/excessive emissions may be lain down in/removed from regions for which little direct constraint is available from the utilised set of observations, but only a more global sensitivity exists in the model (such as the AM region). Maybe not surprisingly these are the regions for which the authors find the strongest changes in a posteriori emissions, a result that somewhat differs from conclusions in previous work. The regional sub-division certainly needs some further justification. This could be done by a more in-depth validation of the model performance at surface sites in contrasting areas like Europe vs. East Asia. For this the use of additional surface observations should be considered (see major comment 3).

We agree that the coarse spatial grouping limits the validity of affects the results to an extent, although it provides more comprehensive information than a similar box model approach. Unfortunately, this approach does assume correlation over large geographical regions that span differing socio-economic regimes. However, aggregation errors such as these are a known drawback of the methodology used in our work, and we discuss this in the main text. To reduce computational cost 5 regions were chosen, although we agree future studies using a similar approach could potentially seek to increase, or vary geographically, the number of regions chosen. The regions were chosen based on socio-economic background but also natural emission regions, and were partially derived by grouping regions of the existing Transcom basis function map (DeFries *et al.*, 1994). Increasing the number of regions used in the inversion would also likely reduce the constraint provided by the limited number of observations used in the study (see next response for more details), which we tried to avoid.

We have now included this caveat in the conclusion and detailed that posterior emissions within a domain are incorrectly assumed to have perfect correlation. We have included that this likely results in a positive bias within a domain being offset by a negative bias elsewhere in the domain. We have referenced our justification for the chosen regions in section 2, with reference to DeFries *et al.* (1994).

As stated correctly on page 2 line 23, Rigby et al. (2017) and Turner et al. (2017) both conclude that the problem of the post 2007 methane rise may be under-constrained using the observed CH4 concentrations, 13CH4/12CH4 ratios and other tracers. Their conclusion is based on simpler box-model simulations without detailed regional division of CH4 emissions. In the present study an even larger number of unknowns is optimised through the inversion. Wouldn't this mean that the individual elements of the state vector are even less well constraint? The authors should spend some time justifying why their more detailed results should be better constrained than those from box-model analyses. In this context it may be worth looking at the covariances in the a posteriori emission and OH factor as well. Large negative covariances may indicate that the inversion cannot clearly distinguish between regions and sectors.

As touched on in the previous response, the number of emission regions and sectors in this study were chosen in order to try to maintain a balance between learning as much as possible about the geographical distribution of the source/sink trends, whilst not under-constraining the posterior solution through use of too large a state vector. In the event, we generally match the number of observations and the number of elements in the state vector quite closely. Our study uses a larger number of regions and observations than previous "box model" studies, along with realistic atmospheric transport representation, and as such provides extra information concerning the distribution of source/sink changes. However, as indicated by our posterior errors and similarly to the other studies mentioned here, the uncertainty of our results is still relatively high, particularly for the OH sink term, and we discuss this in our conclusions.

To investigate this further we have now included some example error posterior correlations, new Figure 11, showing that the error correlations of the off-diagonals is relatively small and therefore, the results are well constrained.

The authors base their inverse flux estimates on a limited set of surface observations (22 flask sampling sites). This may be justified in order to keep the influence of CH4 concentrations to 13CH4/12CH4 ratios similar, with the latter only being available at this limited number of locations. However, for validation purposes there would be many more CH4 observations available worldwide (flask and continuous). These should be evaluated as independent observations as well and may better than GOSAT and TCON observations demonstrate the success of the inverse flux estimate.

We agree that the inclusion of surface site observations for independent validation would improve the evaluation. We have added two independent site validations (new figure 5) for both CH4 and  $\delta^{13}$ CH4. As mentioned, the  $\delta^{13}$ CH4 observations are somewhat limited by the duration of the time series and this has been noted in the updated text.

P2L26f: Although the global a priori emissions by source category are available in Table 1 and regionally divided a posteriori emissions are given in Table 4, I am missing the same kind of information for the a priori. An additional table in the style of Table 4 but for the a priori emissions should be added.

We agree that extra detail on the prior can be added for completeness and have now added this to table 4 and updated the text to include this.

P4L7ff: If I correctly understand the inversion setup, the inversion step is performed on batches of 12 months. Does this mean that the emissions from the previous year are not influenced by the observations of the next year at all? Meaning that January observations will not influence December emissions from the previous year? This would result in December, but probably also November and October, emissions always being constraint by less observations than emissions in other months and, therefore, probably are less corrected from their a priori values and/or show systematically larger a posteriori uncertainties than emissions in other months. Was this observed in the a posteriori factors?

We agree that the nature of the experimental setup results in early year emissions being constrained by more observations then those later in the year. This has been investigated and the posterior error is found to be similar in the early months to the later months. As a result, we do not think the influence of few observations constraining posterior emissions later in the year is noticeable. We have added this to the text and explained that the posterior error for January emissions is similar in magnitude to the error for December emissions.

**P5L16: The wording is not very precise here. J is a cost function and the inversion will find its minimum. J is not a minimisation function. Instead equation 4 represents the analytical minimum of equation 3.**

We agree that the wording is not clear here and have updated the text before equation 3 to indicate that we calculate the cost function to quantify the optimisation and before equation 4 the text now indicates that the minimum is found using the Tarantola and Valette equation.

P5L24: R is not the covariance matrix of the observations alone. R contains the observation/model mismatch covariance. Later this fact is taken care of by adding a model uncertainty to R, but it should be correctly introduced here.

We agree and have added in that R includes both observation and model error.

*P6L6: Was any month-to-month variability of the emissions included in the a priori? If yes where was it taken from?*

We did not include details of this in the original text, but have now corrected the text in page 3 line 25 to comment that prior emissions vary at a monthly timescale.

P6L10f: This is a bit simplistic since the model uncertainty most likely varies with the location of the observation and the question how representative the model grid cell can be for a given site. There have been many different approaches in the past on how to assign site-dependent model uncertainties and, hence, this point should be justified a bit more.

As noted by reviewer 1 this detail was not included in the original submission, we have included a sentence acknowledging that the magnitude of transport uncertainty varies between sites but due to a lack of information we have taken the simple approach and assumed all uncertainties are equal.

P7L6 and elsewhere: A lot of this RMSE is due to a bias in the a priori simulation. It would be better to calculate a bias-corrected RMSE instead. The bias could be mentioned separately. In general it would be nice to include all these comparison statistics in a table as well (in the main text for all discussed inversions and observational data sets and in the supplementary material for all sensitivity inversions).

We agree that the high RMSE values are caused by the bias between the prior and observations. We have now included an explanation of this cause in elevated RMSE. We have also referenced it to Figures 1 to 4, showing the bias against both assimilated and non-assimilated observations. We have used RMSE and not bias-corrected RMSE because the bias is not constant and grows from zero at the start of the simulation. As a result the total offset contributes to the overall error and is reported.

*P7L23f: I don't think it is the model that is growing here. What about 'simulated atmospheric methane growth rates' instead?*

We agree the wording is much clearer as suggested and have made the recommended changes.

P7L27f: This behaviour is very strange. For all other sites an increase in concentrations from a priori to a posteriori simulations was observed. Why not for Garmisch, a central European site not too far away from the Bremen site, where differences in the a priori and a posteriori simulations are as expected? One potential source of mismatch may be the location of Garmisch at the northern edge of the Alps, potentially introducing large mismatches due to smoothed model topography. Still this would not explain the lack of an increase from a priori to a posteriori. Although a detail, this needs to be checked again.

We agree this mismatch is unusual and have since checked the data. We have spotted a coding error that led to this result and have now fixed it to present the actual posterior estimates and have updated the text and plots accordingly.

P8L5 and Figure 5: The estimated a posteriori OH time series should also be compared with work by other authors (e.g. Rigby et al. 2017). If OH is really the main driver of the post 2007 CH4 rise it would be good to know how TOMCAT OH compares to previous work.

We agree that more detailed comparison with other work should be made. We have re-written section 3 and included detailed comparison with both the Rigby et al. and Turner et al. studies.

P8L15: A reference to Table 6 should be added here.

This section has been rewritten, but we have now referenced Table 6 in the equivalent section of the re-written version.

*P9L11: A reference to Figure 5 should be added here.*

This section has been rewritten, but we have now referenced Figure 5 (now 6) in the equivalent section of the re-written version.

*P9L28f:* How similar? These numbers are not given anywhere. One can only guess them from the figures. A table (like Table 4) with the a posteriori emissions for the INVCL case should be provided and the same for all sensitivity inversion (supplement).

We agree a quantitative description of the posterior estimates from the different sensitivities is important, as a result we have included Table 8, which provide these values.

P9L28f: How is the a posteriori performance for this experiment (S4)? Just because one sensitivity run gives different a posteriori emissions it doesn't have to be wrong. But if it also fails to reproduce the observations, then the given conclusion may be correct.

We agree that any single sensitivity test might provide a more realistic representation and therefore should not be discounted just because it is an outlier. Most of the sensitivities provide similar performance when compared with observations; with some exceptions. For S4 we isolate it as an anomaly due to the magnitude of the interannual variability, we consider annual energy sector variability for S4 to be too large to represent a realistic scenario. For example, in 2009 global energy sector emissions are around 3 times higher than the values for other years. We have added in this justification for our conclusion in the text.

P10L31f: This sounds a bit like the authors of Rigby et al. worked on the current study as well. Which is not the case. This work may extend the previous work by using a more complex transport model, but other than that the approaches are fairly different and unrelated (inversion system, used observations, etc.). So I would not write that it extends the work of Rigby or others, but rather it adds to the results gained by others.

We agree the wording is ambiguous and have clarified the text to now indicate that we added to the results from other studies.

P11L7: 'larger errors'. What kind of errors? Needs to be repeated here.

We agree this is also ambiguous, we have commented on the inversion being under constrained and the correlation with observations being reduced.

P11L7f: The sentence 'The constraint improves when the  $\delta$ 13CH4 observations are introduced' should be re-written to be more precise. What about: 'The agreement of the simulations with observations improved when additional  $\delta$ 13CH4 observations are used to constrain CH4 fluxes.'.

We agree the current structure does not explain the improvement, we have modified the text following the suggested re-write.

P11L12: This conclusion is just based on the different trend compared with GOSAT, whereas the trend in surface observations was captured well in the a posteriori simulation. Does that mean that there is a potential trend in the bias between GOSAT and surface observations? Would there be any GOSAT validation studies that may provide some clarification?

We have now extended the validation to apply not only to GOSAT, but also TCCON. This has been added to the results and conclusion, both in the text and figures. From this the posterior underestimates the growth in both GOSAT and TCCON, suggesting that there is no measurement bias in GOSAT. The reason for the bias between surface and column measurements is certainly interesting and might highlight column errors in the posterior, although potential bias in column observations might play a role. We have included this is in the conclusion.

P11L15f: Once again: There are more surface observations available than used in this study. They should be used for validation during this critical period.

We have now included independent surface observations for validation.

P11L29: It is unclear which period is referred to here? Table 5 suggests a growth rate in the energy sector of the AO region of 1.5 Tg yr-2 the text states -2.2 Tg yr-2. What is correct?

The text has now been updated, the text and table provide different values based on different time periods, 2003-2015 and 2003-2012.

P12, 1st paragraph: This section should also repeat what was stated in the introduction concerning previous inverse modelling studies (P2L21ff), especially since the presented results contradict/correct these earlier findings.

This discussion has now moved to the results section and details the comparison with existing studies in more depth to reflect what was in the introduction.

Figure 1: It is impossible to see the red dotted lines in many of the sub-panels (also the ones for  $\delta$ 13CH4). Either the figure needs to be enlarged/split or an additional color and solid line should be used for INV-CH4.

We agree, and the plot has been updated with a new colour for the CH4-only inversion.

Table1, Table4, Table6: These should also contain the uncertainty estimates.

We agree that error values for table 1 and table 4 provide clear information on posterior uncertainties, for table 6 we have included the uncertainties more in the text and figures for clarity as the tables already contain a lot of information.

**Table1: Maybe I missed this before, but does the missing number for the soil sink mean that it was neglected completely? If it was only not-optimised its value should still be part of this table.**

We agree the model soil sink value should be given, we have added it to the table for the prior value; although because it is not optimised in the inversion we do not provide a posterior value.

**Attribution of recent increases in atmospheric methane through 3-D inverse modelling**

Joe McNorton1,2, Chris Wilson1,3, Manuel Gloor4, Rob Parker3,5, Hartmut Boesch3,5, Wuhu Feng1,6, Ryan Hossaini7, Martyn Chipperfield1,3

1. School of Earth and Environment, University of Leeds, Leeds, UK.

2 Research Department, European Centre for Medium-Range Weather Forecasts, Reading, UK.

3. National Centre for Earth Observation, University of Leeds, Leeds, UK.

4 School of Geography, University of Leeds, Leeds, UK.

5 Earth Observation Science Group, Department of Physics and Astronomy, University of Leicester, Leicester, UK. 6National Centre for Atmospheric Science, University of Leeds, Leeds, UK. 7Lancaster Environment Centre, Lancaster University, Lancaster UK.

Correspondence to: Joe McNorton (Joe.McNorton@ecmwf.int)

**Abstract.**

5

- 15 The atmospheric methane (CH4) growth rate has varied considerably in recent decades. Unexplained renewed growth after 2006 followed seven years of stagnation and coincided with an isotopic trend toward CH4 more depleted in 13C, suggesting changes in sources and/or sinks. Using surface observations of both CH4 and the isotopologue ratio value ( $\delta^{13}$ CH4) to constrain a global 3D\_3-D chemical transport model (CTM), we have performed a synthesis inversion for source and sink attribution. Our method extends on previous studies by providing monthly and regional attribution of emissions from 6 different sectors
- 20 and changes in atmospheric sinks for the extended 2003-2015 period. Regional evaluation of the model CH4 tracer with independent column observations from the Greenhouse gases Observing SATellite (GOSAT) shows improved performance when using posterior fluxes (R = 0.94-0.96, RMSE = 8.3-16.5 ppb), relative to prior fluxes (R = 0.60-0.92, RMSE = 48.6-64.6 ppb). Further independent validation with data from the Total Carbon Column Observing Network (TCCON) shows a similar improvement in the posterior fluxes (R = 0.990.87, RMSE = 18.821.4 ppb) compared to the prior (R = 0.710.69, RMSE = 18.821.4 ppb) compared to the prior (R = 0.710.69, RMSE = 18.821.4 ppb) compared to the prior (R = 0.710.69, RMSE = 18.821.4 ppb) compared to the prior (R = 0.710.69, RMSE = 18.821.4 ppb) compared to the prior (R = 0.710.69, RMSE = 18.821.4 ppb) compared to the prior (R = 0.710.69, RMSE = 18.821.4 ppb) compared to the prior (R = 0.710.69, RMSE = 18.821.4 ppb) compared to the prior (R = 0.710.69, RMSE = 18.821.4 ppb) compared to the prior (R = 0.710.69, RMSE = 18.821.4 ppb) compared to the prior (R = 0.710.69, RMSE = 18.821.4 ppb) compared to the prior (R = 0.710.69, RMSE = 18.821.4 ppb) compared to the prior (R = 0.710.69, RMSE = 18.821.4 ppb) compared to the prior (R = 0.710.69, RMSE = 18.821.4 ppb) compared to the prior (R = 0.710.69, RMSE = 18.821.4 ppb) compared to the prior (R = 0.710.69, RMSE = 18.821.4 ppb) compared to the prior (R = 0.710.69, RMSE = 18.821.4 ppb) compared to the prior (R = 0.710.69, RMSE = 18.821.4 ppb) compared to the prior (R = 0.710.69, RMSE = 18.821.4 ppb) compared to the prior (R = 0.710.69 ppc) compared to the prior (
- 25 55.39 ppb). Based on these improved posterior fluxes, the inversion results suggest the most likely cause of the renewed methane growth is a post-20076 1.8±0.4% decrease in mean OH, a 12.9±2.7% increase in energy sector emissions, mainly from Africa/Middle East and Southern Asia/Oceania, and a 2.6±1.8% increase in wetland emissions, mainly from Northern Eurasia. The posterior wetland increases are in general agreement with bottom-up estimates, but the energy sector growth is greater than estimated by bottom-up methods. The model results are consistent across a range of sensitivity analyses performed.
- 30 When forced to assume a constant (annually repeating) OH distribution, the inversion requires a greater increase in energy sector (13.6±2.7%) and wetland (3.6±1.8%) emissions and but also introduces an 11.5±3.8% decrease in biomass burning emissions. Assuming no prior trend in sources and sinks slightly reduces the posterior growth rate in energy sector and wetland emissions, and further increases the amplmagnitude of the negative OH trend. We find that possible tropospheric Cl variations do not to influence δ13CH4 and CH4 trends, although we suggest further work on Cl variability is required to fully diagnose

this contribution. While the study provides quantitative insight into possible emissions variations which may explain the observed trends, uncertainty in prior source and sink estimates and a paucity of  $\delta^{13}$ CH4 observations limit the accuracy robustness of the posterior estimates.

**1 Introduction**

- 5 The atmospheric concentration of methane (CH4) has been increasing globally since 2007, following a slowdown in growth from 1999 to 2006 (Dlugokencky *et al.*, 2017). The onset of the observed increase in CH4 coincides with an isotopic trend to lighter CH4, more depleted in 13C (Nisbet *et al.*, 2014). The 13CH4:12CH4 ratio (denoted by the δ13CH4 
[revised manuscript text omitted]
_{0H} \frac{\Delta \varphi_{\mathbf{x}}}{\Delta x_{0H}}(l, t) + x_{ini}\varphi_{ini}(l)$$
(1)

$$\psi(\mathbf{x},l,t) = \sum_{s=1}^{n_{source}} \sum_{i=1}^{n_{reg}} \sum_{m=1}^{n_{month}} x_{i,m,s} \frac{\Delta \psi}{\Delta x_{i,m,s}} (l,t) + x_{OH} \frac{\Delta \psi}{\Delta x_{OH}} (l,t) + \psi_{ini}(l)$$
(2)

20 Note that we use Δ here to represent change, in order to avoid confusion with the isotopologue δ13CH4. Basis functions Δψ/Δxi,m,s and Δψ/Δxi,m,s are sensitivities of atmospheric CH4 and δ13CH4 at a particular time and location to an emission of 1 Tg of CH4 from a region *i* during a particular month *m*, for an emission sector *s*. Each *xi,m,s* is a scaling factor applied to the contribution from each basis function, and is initially set equal to the prior value of the emission. Similarly, Δψ/Δx0H and Δψ/Δx0H are the sensitivities of the mixing ratio and δ13CH4 at a measurement location to a change in the global OH concentration, linearised around the prior, and *x0H* is initially set to be the prior OH concentration. *xini* is a dimensionless scaling factor initially set to be 1. Although the emissions in each region and source type are split into 12CH4 and 13CH4, the relative emissions of each isotopologue from each region for each source type are not included as separate basis functions. The 'state vector' *x* comprises of the individual emission scaling factors *xi,m,s*, for all *i, m* and *s*, along with *x0H* and *xini*. Sensitivity experiments performed for tropospheric Cl follow the same formulation with Cl terms replacing OH terms.

10

Varying atmospheric CH4 concentrations in the inversions should in principle results in a non-linear feedback on OH concentration. This feedback is not accounted for and does not influencein the offline OH field used in our inversion. To resolve this, an online OH field could in principle be used with an iterative minimization of the cost function. However, showever, Bousquet *et al.* (2011) found that the variationsmall variation in CH4 concentration between the prior and posterior is relatively small and assumed to have had a negligible influence on OH concentration.

The model OH is constrained by CH4 and δ13CH4 andbut not by other species, such as methyl-chloroform (MCF). MCF was excluded because of uncertainty in emissions and a diminishing concentration (<5 ppt), particularly during the later period of the study (Liang *et al.*, 2017). Due to the large uncertainty relative to the observed MCF concentrations in this period, including the extra species within the inversion would not add any extra constraint on the global OH concentration.

Independent inversions (INV-FULL) were performed for each year from 2003 to 2015. Initial conditions for each year are provided by a forward simulation for the previous year driven by derived posterior emissions and loss rates, with 2003 initial
conditions taken from a 2002 spin-up inversion. To quantify the optimisation of the flux terms in each region and the sink term, we calculate the cost function, J: To estimate the flux contributions from each region we apply a minimisation function, which calculates the cost function, J:

$$J(\mathbf{x}) = \frac{1}{2} (\mathbf{x} - \mathbf{x}^{b})^{T} \cdot \mathbf{B}^{-1} \cdot (\mathbf{x} - \mathbf{x}^{b}) + \frac{1}{2} (\mathbf{y} - \mathbf{G} \cdot \mathbf{x})^{T} \cdot \mathbf{R}^{-1} \cdot (\mathbf{y} - \mathbf{G} \cdot \mathbf{x})$$
(3)

[revised manuscript text omitted]

---

## Author Response (AR2)

We would like to thank the reviewer for his/her further helpful comments. These are repeated below (in *italics*) followed by our responses.

**Reviewer**

5 This is my second review of the manuscript and there are still a few minor points I would like to see addressed before publication.

Main comments:

*The authors have considerably revised the presentation of their results since the first submission, especially they have followed the suggestions made by Reviewer 1 in restructuring the discussion of the main results (old sections*
10 *3.2 and 3.3). The new presentation is more concise and better structured than before.*

*Concerning my major concerns from the previous review the authors have included two additional figures and discussion to address the question of independent surface measurements for validation (new Fig 5) and posterior covariance (new Fig 11). The latter point is satisfactorily answered by the new figure and corresponding discussion, whereas for the first point some minor questions remain (see below).*

15 *However, my most major concern (the definition of the emission regions) still requires some additional justification and discussion. The authors mention in the reply and added to the manuscript that the region definition follows 'existing Transcom basic functions' and they cite DeFries et al. (1994). However, DeFries et al. (1994) presents a satellite based global vegetation classification, which may be relevant for defining CO2 emission/uptake regions, but has very little to do with CH4 emissions. Maybe the definition of Transcom regions was based on DeFries, but*
20 *I still don't see the relevance for CH4. My major concern of considerably different economic developments in the large Eurasian region is not considered in any detail. Next to the (in my view insufficient) motivation of the region definition the authors have added a note of caution in the conclusions. However, the authors seem to suggest that the kind of aggregation errors they are dealing with will level out when looking at a region total (somewhere too high, somewhere too low). That this does not have to be the case and that it is related to where observations are*
25 *available was nicely demonstrated by Kaminiski et al. (2001). Although their findings are based on much coarser transport simulations their regions are similarly coarse as in the current study. In conclusion, I would like to ask the authors for a more meaningful motivation of the region definition (beside the DeFries) reference and change the comment in the conclusion towards the possibility of the aggregation error causing significant biases in the posterior fluxes.*

30 We have added further justification in section 2.1.1 describing the subjective nature of aggregating by both socio-economic and biome considerations. As suggested by the reviewer, we have now used the Eurasian region example to highlight that whilst there are socio-economic differences within the region it typically represents boreal and temperate biomes, which are important when considering natural emissions (wetlands and biomass burning). The Asian region below, whilst socio-economically similar the northern region, represents a more
35 tropical biome. We believe for this reason the link to the vegetation structure defined by DeFries et al. is a reasonable choice for $CH_4$ as well as for $CO_2$. There are limited continuous east Eurasian observations with which to perform validation, however aggregated GOSAT comparisons (figure 2) highlight it as the region with the lowest uncertainties in the posterior. We agree however that the regional aggregation is somewhat subjective and

different aggregations may produce different results as suggested by Kaminski et al. (2001). Thus we explain in our revised manuscript that the resolution of the regional aggregation is limited in our study because of the computational cost of performing a synthesis inversion with individual sectors and at a monthly frequency (which already requires 730 3-D model tracers), so a trade-off is necessary. We have also included a statement / caveat in our discussion of the results that the influence of the choice of regional aggregation on posterior fluxes may have biased some of the results and cite the Kaminski *et al.* 2001 paper.

*Other comments - page and line numbers referring to the track-changes version of the manuscript:*

*Page 3, Line 30: Abbreviation for 'kinetic fractionation' is not very commonly used and only twice more in the paper, at which point one does not remember the definition. Please remove the abbreviation and replace by complete term.*

We agree that abbreviating the term makes it more difficult to follow and have updated the abbreviations to provide 'kinetic fractionation' in full.

*Equation 3 (and elsewhere): I don't think the dots are the correct notation for a matrix multiplication here. Also follow the ACP guide on which font type to use for matrices and vectors.*

The notation has been updated and the dots have been removed to follow standard ACP conventions.

*P7, L1f: What is the rational for choosing these sites? Especially the high-altitude site seems to cause more discussions than it can convince that the simulations worked well. Why not use other continuous CH4 surface observations even though no d13CH4 observations are available at such sites? In the end, GOSAT and TCCON don't provide isotopic information either.*

The validation sites were selected from the remaining long-term flask measurements sites that were still available and which had not too many gaps. The chosen sites were also sufficiently far away from assimilated sites to be considered to provide independent/ uncorrelated information.

Based on the reviewer's comment we have been able to identify an issue in previous versions of this paper with the processing of HAGCOC model data related to the height interpolation. The previous plot used incorrect model levels. We have now fixed this and updated the plot. In addition to this we have updated the text to reflect the updated results.

*Figure 5: Why is there a jump in the prior at the end of 2013? This is visible at both locations but not for any of the other sites displayed in Figure 1. It is indeed strange that d13CH4 is decreased so much at HAGCOC. Is there a problem with comparing the simulations with high-altitude data? How was the difference in topography between model and reality considered?*

A similar issue to that commented on above has been found in the processing code. As a result we have redone the comparison of the independent surface sites using the correct model data. The updated plots show improved a posteriori model – data agreement at both stations, as expected. For differences in topography the model height at the observation location is used. We thank the reviewer for prompting us to look at this.

*Figure 11: This is an interesting and helpful figure. There is just one clarification I would like to ask for. Since values in the figure are between -1 and 1, I assume this is not the posterior covariance matrix but some normalised version of it. Please indicate how it was normalised. Furthermore, it is very difficult to interpret the by-sector part of the*

*matrix because no x axis labels are included. I suggest to include a numbering in the existing y axis labels and use the same numbers on the x axis as well. This would allow for a quick identification of which sectors/regions actually showed considerable posterior covariance*

The values shown represent the posterior correlations between the months, sectors and regions, rather than the posterior covariances in order to make them easier to understand. To derive these correlations, the posterior covariances produced using Eqn. 5 have been normalised using the corresponding posterior standard deviations. We have added this detail into the text and Figure 11 as suggested and described how the values were calculated.

We agree that the sector-by-sector interpretation is difficult, as a result we have recreated the figure with more appropriate gridding and labelled the x-axis for the top panel.

References:

[revised manuscript text omitted]